# Formation of a giant unilocular vacuole via macropinocytosis-like process confers anoikis resistance

Jeongsik Kim[1†], Dahyun Kim[1†], Dong-Kyun Kim[1], Sang-Hee Lee[2], Wonyul Jang[1,3*], Dae-Sik Lim[1*]

[1]Department of Biological Sciences, KAIST Stem Cell Center, Korea Advanced Institute of Science and Technology (KAIST), Daejeon, Republic of Korea; [2]Center for Research Equipment, Korea Basic Science Institute, Ochang, Cheongju, Republic of Korea; [3]School of Biological Sciences, Seoul National University, Seoul, Republic of Korea

*For correspondence:
wonyuljang@snu.ac.kr (WJ);
daesiklim@kaist.ac.kr (D-SikL)

[†]These authors contributed equally to this work

Competing interest: The authors declare that no competing interests exist.

## eLife assessment

This **important** study reports the formation of a new organelle, called giant unilocular vacuole (GUVac), in mammary epithelial cells through a macropinocytosis-like process. The evidence supporting conclusions is **convincing**, using state-of-the-art cell biology techniques. This work will be of interest to cell biologists and contribute to the understanding of cell survival mechanisms against anoikis.

**Abstract** Cell survival in metazoans depends on cell attachment to the extracellular matrix (ECM) or to neighboring cells. Loss of such attachment triggers a type of programmed cell death known as anoikis, the acquisition of resistance to which is a key step in cancer development. The mechanisms underlying anoikis resistance remain unclear, however. The intracellular F-actin cytoskeleton plays a key role in sensing the loss of cell–ECM attachment, but how its disruption affects cell fate during such stress is not well understood. Here, we reveal a cell survival strategy characterized by the formation of a giant unilocular vacuole (GUVac) in the cytoplasm of the cells whose actin cytoskeleton is disrupted during loss of matrix attachment. Time-lapse imaging and electron microscopy showed that large vacuoles with a diameter of >500 nm accumulated early after inhibition of actin polymerization in cells in suspension culture, and that these vacuoles subsequently coalesced to form a GUVac. GUVac formation was found to result from a variation of a macropinocytosis-like process, characterized by the presence of inwardly curved membrane invaginations. This phenomenon relies on both F-actin depolymerization and the recruitment of septin proteins for micron-sized plasma membrane invagination. The vacuole fusion step during GUVac formation requires PI(3)P produced by VPS34 and PI3K-C2α on the surface of vacuoles. Furthermore, its induction after loss of matrix attachment conferred anoikis resistance. Our results thus show that the formation of a previously unrecognized organelle promotes cell survival in the face of altered actin and matrix environments.

## Introduction

Cells of higher organisms contain various functional organelles with characteristic morphologies. These organelles compartmentalize the cell interior and are dynamically remodeled or newly formed as adaptive responses to changes in the cellular environment (*Jang et al., 2022*; *Scott et al., 2004*).

The formation of large intracellular vacuoles has been observed across diverse eukaryotic species. Most plant cells, for example, contain a large acidic vacuole that occupies most of the cell volume and functions to maintain turgor pressure (*Taiz, 1992*). Large vacuoles have also been described in mammalian cells. In the developing vascular system, for instance, the emergence of a sizeable intracellular vacuole is thought to underlie endothelial lumenization (*Kamei et al., 2006*). Furthermore, glioblastoma cells expressing oncogenic Ras protein manifest increased macropinocytosis and accumulation of large cytoplasmic vacuoles that result in a nonapoptotic form of cell death known as methuosis (*Maltese and Overmeyer, 2014*; *Overmeyer et al., 2008*). White adipocytes in fat tissue also form a large unilocular sac (lipid droplet) that occupies most of the cytoplasm but contains lipids rather than extracellular fluid. However, whether the ability to form large vacuoles is a generally conserved functional program across mammalian cell types has remained unknown.

Loss of extracellular matrix (ECM) attachment in normal epithelial cells can induce a form of programmed cell death known as anoikis (*Frisch and Francis, 1994*). However, metastatic tumor cells occasionally circumvent such death induction by cell–cell adhesion and cell clustering (*Labuschagne et al., 2019*). Both cell–ECM interaction and cell–cell adhesion directly affect the actin cytoskeleton via integrin or cadherin proteins, respectively (*Bachir et al., 2017*; *Bertocchi et al., 2017*). However, whether extracellular information transmitted to the cell interior via the actin cytoskeleton determines cell fate, such as cell survival or death, has been unclear.

We now show that, in normal epithelial cells that are highly sensitive to the loss of matrix attachment (*Debnath et al., 2002*; *Schafer et al., 2009*), acute disruption of the actin cytoskeleton in matrix-detached cells promotes cell survival rather than cell death (i.e., anoikis resistance). This survival is associated with the emergence of a giant unilocular vacuole (GUVac) that occupies most of the cell volume. The formation of this giant organelle was found to depend on a previously undescribed form of macropinocytosis that requires actin depolymerization and septin recruitment for micron-sized inward plasma membrane curvature. Genetic suppression of GUVac formation in matrix-detached cells led to cell death in response to inhibition of actin polymerization, suggesting that GUVac formation is essential for cell survival in an altered matrix environment. Our findings have thus uncovered a cellular strategy based on the formation of a unique type of organelle to ensure cell survival under conditions of cellular stress.

## Results

### Formation of GUVacs induced by F-actin disruption in epithelial cells originating from secretory tissues in the absence of matrix attachment

Detachment of MCF-10A human mammary epithelial cells from ECM has previously been shown to induce entosis, a nonapoptotic form of cell death initiated by the invasion of one cell by another and resulting in the formation of an entotic vacuole (*Overholtzer et al., 2007*). While exploring this phenomenon, we occasionally found an unusual type of cell that contained a single GUVac that occupied almost the entire cytoplasm (*Figure 1A and B*). These GUVacs differed from entotic vacuoles in that they did not contain cells (i.e., no cell-in-cell structures were apparent). Given that ECM detachment results in a gradual change in the actomyosin cytoskeleton, we tested whether changes to either actin or myosin might be associated with GUVac formation. Treatment with latrunculin B (LatB) or cytochalasin D (CytoD), inhibitors of F-actin polymerization, induced a gradual increase in the number of cells with a GUVac during suspension in both MCF10A and human primary mammary epithelial cells (HMEpiCs) (*Figure 1C–E*, *Figure 1—figure supplement 1A*). In contrast, treatment with an actin stabilizer (jasplakinolide), an inhibitor of microtubule polymerization (nocodazole), a myosin inhibitor (blebbistatin), or inhibitors of Rho, the Rho kinase ROCK (Y-27632), Rac (EHT 1864), or Cdc42 (ML 141) had no such effect (*Figure 1C–F*). To further demonstrate the effect of actin depolymerization on GUVac formation, we overexpressed the SpvB (*Harterink et al., 2017*), a toxin derived from *Salmonella enterica* that can cause complete loss of actin filaments in cells and observed similar phenotype to cells treated with LatB (*Figure 1G and H*), confirming that actin depolymerization during the loss of matrix attachment promotes GUVac formation. To investigate the potential of generating GUVac in response to actin stress across various cell lines, we examined established cell lines derived from multiple tissues. Interestingly, we noted that cells originating from secretory tissues, such as gastric (HFE-145, AGS, MKN-45), thyroid (BCPAP), and pancreatic (PANC-1) tissues, displayed a tendency

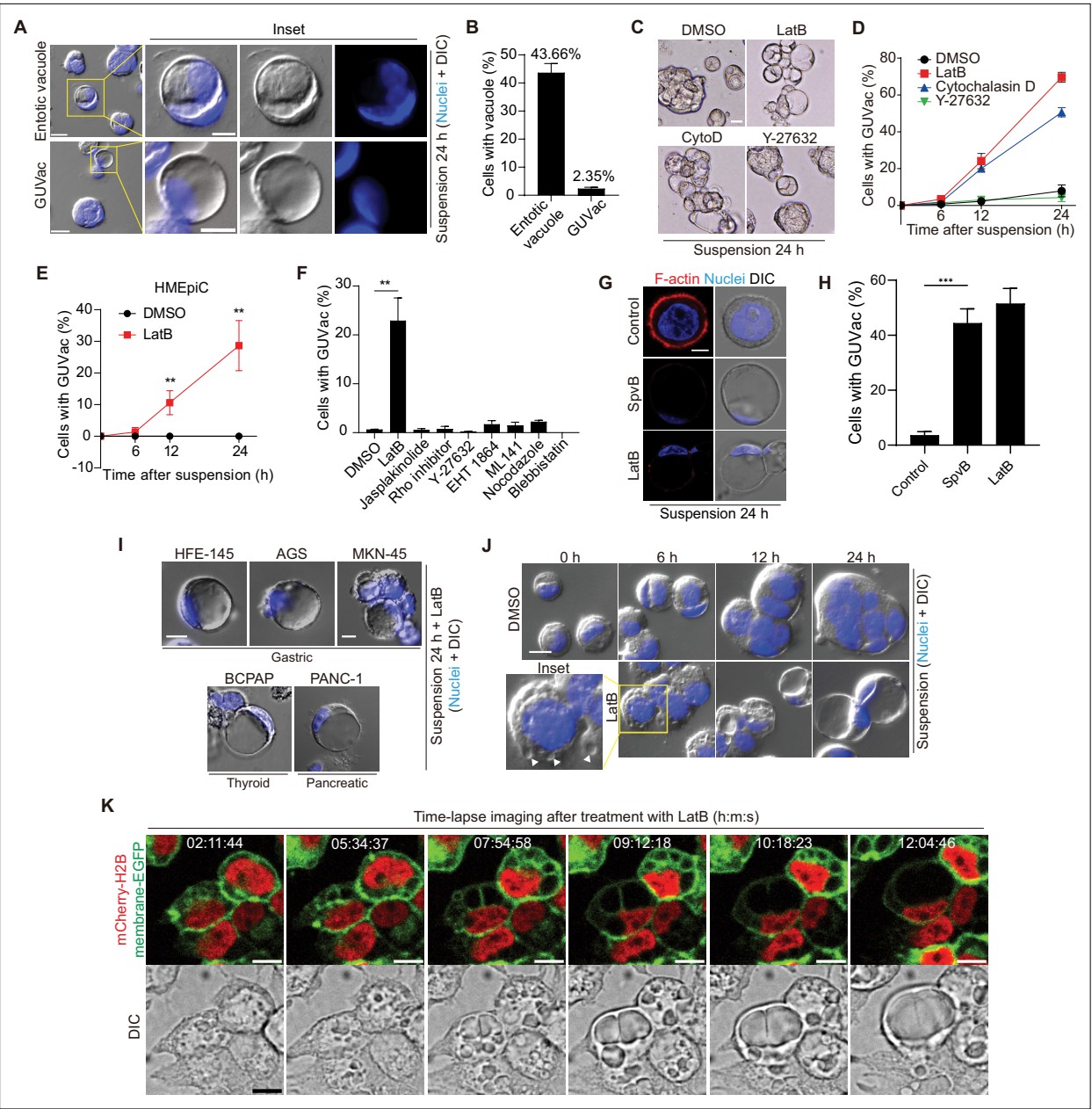

**Figure 1.** Giant unilocular vacuole (GUVac) formation induced by F-actin disruption in matrix-deprived mammary epithelial cells. (**A**) Representative differential interference contrast (DIC) microscopy images of suspended MCF-10A cells showing entotic vacuole or GUVac formation after culture for 24 hr. Nuclei were stained with Hoechst 33342. Scale bars: 20 µm (main panels) or 10 µm (inset). (**B**) Percentage of suspended MCF-10A cells showing entotic vacuole or GUVac formation after 24 hr (n = 772). (**C**) Representative bright-field microscopy images of MCF-10A cells suspended with F-actin cytoskeleton inhibitors. Scale bar: 15 µm (**D**) Percentage of suspended MCF-10A cells showing GUVac formation after exposure to the indicated drugs for the indicated times. dimethyl sulfoxide (DMSO): 0 hr (n = 512), 6 hr (n = 723), 12 hr (n = 690), and 24 hr (n = 690). latrunculin B (LatB): 6 hr (n = 634), 12 hr (n = 693), and 24 hr (n = 428). Cytochalasin D: 6 hr (n = 613), 12 hr (n = 618), and 24 hr (n = 464). Y-27632: 6 hr (n = 448), 12 hr (n = 601), and 24 hr (n = 660). (**E**) Percentage of suspended human primary mammary epithelial cells (HMEpiCs) showing GUVac formation after incubation with DMSO or LatB for the indicated times. DMSO: 6 hr (n = 722), 12 hr (n = 647), 24 hr (n = 417). LatB: 6 hr (n = 1225), 12 hr (n = 1505), and 24 hr (n = 1335). (**F**) Percentage of suspended MCF-10A cells showing GUVac formation after incubation with the indicated drugs for 18 hr. DMSO (n = 1087), LatB (n = 1634), jasplakinolide (n = 2578), Rho inhibitor (n = 839), Y-27632 (n = 1322), EHT 1864 (n = 905), ML 141 (n = 997), nocodazole (n = 1342), and blebbistatin (n = 520). (**G**) Representative DIC microscopy images with phalloidin staining show disruption of the actin cytoskeleton and the GUVac formation in SpvB-expressing cells after suspension culture for 24 hr. Scale bar: 10 µm (**H**) Percentage of GUVac formation in control or SpvB-expressing MCF-10A cells. (**I**) Representative DIC microscopy images after suspension culture of indicated cell lines for 24 hr with LatB. Scale bar: 10 µm (**J**) Representative DIC images of suspended MCF-10A cells treated with DMSO or LatB for the indicated times. White arrowheads indicate the accumulation of vacuoles. Nuclei were stained with Hoechst 33342. Scale bar, 20 µm. (**K**) Representative fluorescence and DIC time-lapse images of MCF-10A cells

*Figure 1 continued on next page*

*Figure 1 continued*

expressing mCherry-H2B and membrane-targeted EGFP obtained at the indicated times after the onset of LatB treatment. Times are presented as hour:minute:second (h:m:s). Scale bars, 10 μm. All quantitative data are means ± SD. The n values represent the total number of cells quantified for two (**D**) or three (**B, E, F, H**) independent experiments. \*\*p<0.01 (two-tailed unpaired *t*-test) for the indicated comparison (**F, H**) or versus the corresponding value for DMSO (**E**).

The online version of this article includes the following source data and figure supplement(s) for figure 1:

**Source data 1.** Quantification data corresponding to *Figure 1B*.

**Source data 2.** Quantification data corresponding to *Figure 1E*.

**Source data 3.** Quantification data corresponding to *Figure 1F*.

**Source data 4.** Quantification data corresponding to *Figure 1H*.

**Figure supplement 1.** GUVac develops specifically in suspended cells that originate from secretory-related tissues.

**Figure supplement 1—source data 1.** Quantification data corresponding to *Figure 1—figure supplement 1D*.

**Figure supplement 2.** GUVac membranes do not consistently align with particular organelle markers.

to form GUVac, similar to what was observed in mammary MCF10A and HMEpic cells (*Figure 1C, E, and I*). In contrast, cell types from other tissues did not exhibit this phenomenon (*Figure 1—figure supplement 1A and B*). These results suggested that the development of GUVac might be an intrinsic adaptive program specific to epithelial cells derived from secretory tissues.

To obtain further insight into the process of GUVac formation, we carefully examined suspended cells collected at various times after exposure to LatB. At 6 hr after the onset of LatB treatment, suspended cells already contained several indentations resembling nascent vacuoles at the cell surface, whereas cells treated with dimethyl sulfoxide (DMSO) vehicle did not (*Figure 1J*). Examination of the cells at 12 and 24 hr revealed that the resulting vacuoles gradually increased in size and fused with each other, eventually giving rise to one giant vacuole (*Figure 1J*). We validated these observations by live imaging of suspended MCF-10A cells expressing both mCherry-H2B, which labels nuclei, and plasma membrane-targeted enhanced green fluorescent protein (EGFP). Time-lapse images clearly confirmed that individual nascent vacuoles that arose at ~2 hr after exposure to LatB became enlarged and eventually merged into one GUVac (*Figure 1K*), consistent with the analysis of fixed samples. Adherent cells or suspended cells incubated with reconstituted basement membrane (Matrigel) did not form GUVacs under treatment with LatB or cytochalasin D, suggesting that matrix attachment suppresses GUVac formation (*Figure 1—figure supplement 1C and D*).

Next, we attempted to determine whether GUVac displays markers of other endomembrane systems. At early time points (1 hr), we observed several large vesicles that had taken up 70 kDa dextran and exhibited EEA1 or Rab5, markers of early endosomes. By 6 hr, some of these large vesicles showed LysoTracker positivity, indicating a transition from early to late endosomal fate, similar to the maturation process of conventional macropinocytic vesicles (*Figure 1—figure supplement 2A*). However, once the vesicles fused, grew, and became GUVac, these markers did not consistently correspond with the GUVac membrane but were instead unevenly distributed around it (*Figure 1—figure supplement 2B and C*). This made it difficult to determine whether they were localized to separate organelles or part of the GUVac membrane. Interestingly, we found that the Transferrin receptor (TfR), which also marks a general membrane population involved in the endocytic pathway (such as PM invagination), was evenly distributed within the GUVac membrane (*Figure 1—figure supplement 2B and C*). Therefore, GUVac appears to possess heterogeneous characteristics of the endocytic membrane, mainly with the TfR marker (likely due to PM invagination) and some partial endomembrane system markers.

## Macropinocytosis-like process contributes to GUVac formation

To further characterize the mechanism of GUVac biogenesis, we next applied transmission electron microscopy (TEM) to examine the ultrastructure of matrix-detached MCF-10A cells at various times after exposure to LatB. At 6 hr, we detected inwardly curved structures at the plasma membrane in LatB-treated cells but not in DMSO-treated cells (*Figure 2A and B*). The diameter of these endocytic structures was >500 nm, similar to that of macropinocytosis-derived endosomes (*Swanson, 2008*). To examine whether GUVac formation is the result of macropinocytosis, we tested the effect of 5-(N-ethyl-N-isopropyl) amiloride (EIPA), a Na⁺/H⁺ exchanger inhibitor that is widely applied to

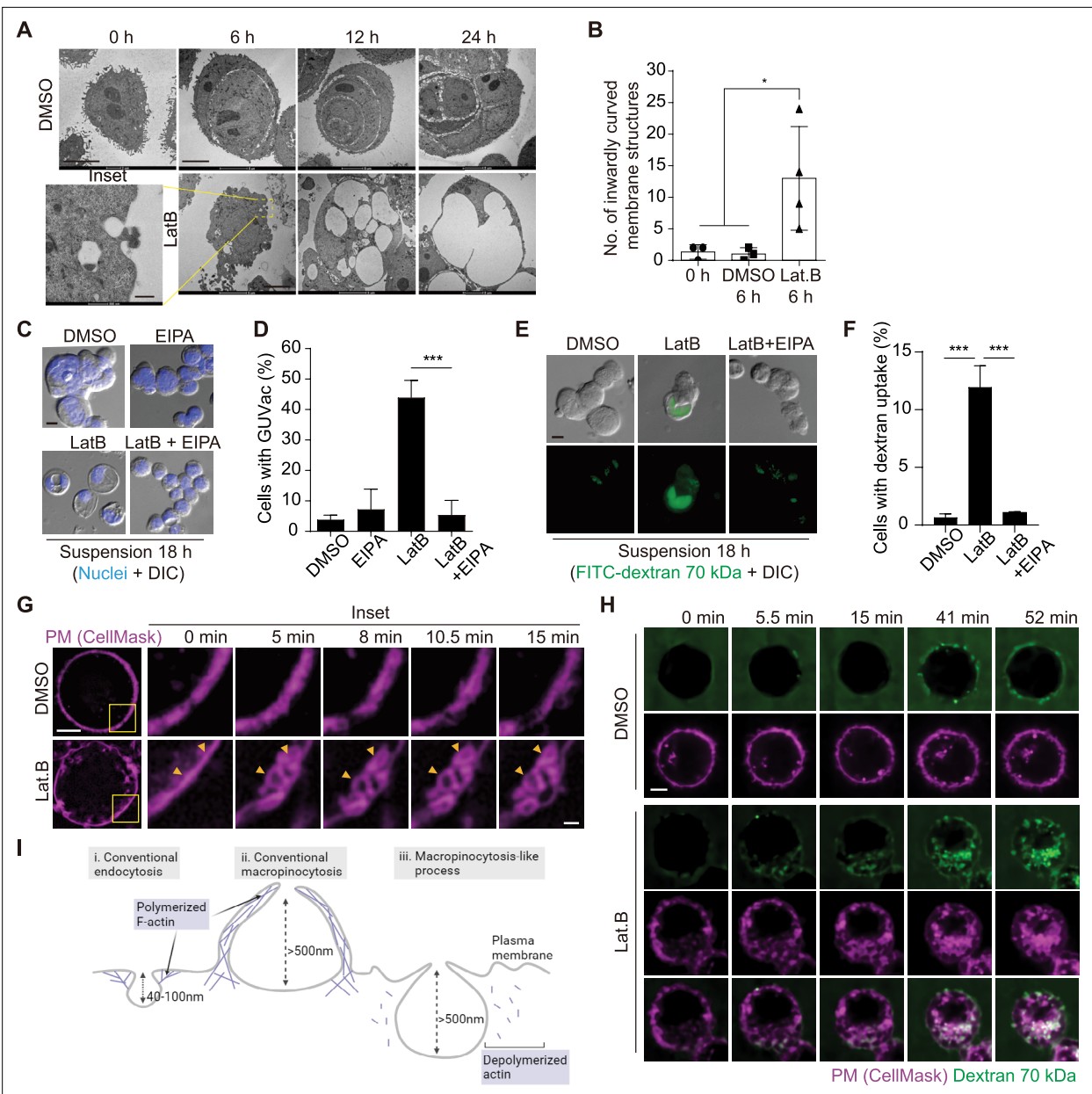

**Figure 2.** Macropinocytosis-like process contributes to giant unilocular vacuole (GUVac) formation. (**A**) Representative transmission electron microscopy (TEM) images of suspended MCF-10A cells treated with dimethyl sulfoxide (DMSO) or latrunculin B (LatB) for the indicated times. Scale bars, 5 µm (main panels) or 500 nm (inset). (**B**) Number of inwardly curved plasma membrane structures with a diameter of >500 nm detected by TEM in individual suspended MCF-10A cells treated with DMSO or LatB for 6 hr. Each data point represents an individual cell analyzed. (**C**) Representative differential interference contrast (DIC) images of suspended MCF-10A cells treated with 5-(*N*-ethyl-*N*-isopropyl) amiloride (EIPA) or LatB for 18 hr. Nuclei were stained with Hoechst 33342. Scale bar, 20 µm. (**D**) Percentage of cells showing GUVac formation in experiments as in (**C**). DMSO (n = 1489), EIPA (n = 1256), LatB (n = 1482), and LatB + EIPA (n = 1159). (**E**) Representative DIC and FITC-dextran fluorescence images of suspended MCF-10A cells treated with LatB or EIPA for 18 hr. The cells were exposed to FITC–dextran (70 kDa) at 1 mg/ml. Scale bar, 20 µm. (**F**) Percentage of cells showing FITC-dextran uptake in experiments as in (**E**). DMSO (n = 797), LatB (n = 1534), and LatB + EIPA (n = 821). (**G**) Representative super-resolution structured illumination microscopy (SIM) time-lapse images of suspended MCF-10A cells showing the plasma membrane labeled by CellMask obtained at the indicated times after DMSO or LatB treatment. Arrowheads indicate plasma membrane invagination and vesicle formation. Scale bars, 5 µm (main panels) or 1 µm (inset). (**H**) Representative spinning disk confocal time-lapse images of suspended MCF-10A cells showing CellMask-labeled plasma membrane and 70 kDa dextran obtained at the indicated times after DMSO or LatB treatment. Scale bar, 5 µm. (**I**) Schematic showing the differences among conventional endocytosis, conventional macropinocytosis, and unconventional macropinocytosis-like process with regard to the size of endocytic cups and reliance on actin polymerization. Schematic was created using Biorender. All quantitative data are means ± SD. The *n* values represent the total

*Figure 2 continued on next page*

*Figure 2 continued*

number of cells examined in three independent experiments (**D, F**). *p<0.05, ****p<0.0001 by one-way ANOVA (**B**) or one-way ANOVA with Tukey's multiple comparisons (**D, F**).

The online version of this article includes the following source data and figure supplement(s) for figure 2:

**Source data 1.** Quantification data corresponding to *Figure 2B*.

**Source data 2.** Quantification data corresponding to *Figure 2D*.

**Source data 3.** Quantification data corresponding to *Figure 2F*.

**Figure supplement 1.** Colocalization of macropinosomes with the invaginated plasma membrane during early vacuole formation.

block macropinocytosis (*Koivusalo et al., 2010*). Treatment with EIPA resulted in significant inhibition of GUVac formation in suspended cells treated with LatB (*Figure 2C and D*). We also observed that LatB-treated suspended cells engulfed fluorescein isothiocyanate (FITC)-labeled high-molecular-weight dextran (70 kDa), which is not capable of being taken up by micropinocytosis (*Li et al., 2015*). This engulfment was inhibited by EIPA (*Figure 2E and F*). We next utilized super-resolution structured illumination microscopy (SIM) to examine this phenomenon in live cells at earlier time points following LatB treatment. We found that a portion of the plasma membrane stained with CellMask showed signs of invagination with a diameter of approximately 1 μm. Within 1 hr after LatB treatment, these invaginations eventually pinched away from the cell surface. In contrast, control cells did not exhibit any inward plasma membrane invaginations (*Figure 2G*, *Video 1*, and *Figure 2—figure supplement 1A*). The live super-resolution imaging results strongly indicate that the formation of large vesicles (0.5–1 μm in diameter) primarily originates from the invaginated plasma membrane. Furthermore, we visualized the live uptake of high-molecular-weight dextran (70 kDa) in conjunction with CellMask plasma membrane staining in suspended cells immediately after LatB treatment. We observed the gradual accumulation of dextran-positive macropinosomes in LatB-treated cells, but such accumulation was not observed in control cells. These macropinosomes were found to be colocalized with the invaginated plasma membrane, as revealed by live spinning-disk confocal and super-resolution SIM images (*Figure 2H*, *Video 2*, and *Figure 2—figure supplement 1B*). Collectively, these results suggest that the initial formation of these large vesicles results from the internalization of substantial portions of the plasma membrane. This distinctive form of macropinocytosis-like phenomenon responsible for GUVac formation is unique in that it relies on actin depolymerization and exhibits inwardly curved plasma membrane structures, in contrast to the conventional macropinocytosis, which entails actin polymerization and the creation of outward membrane protrusions for engulfing extracellular fluid (*Swanson, 2008*; *Figure 2I*).

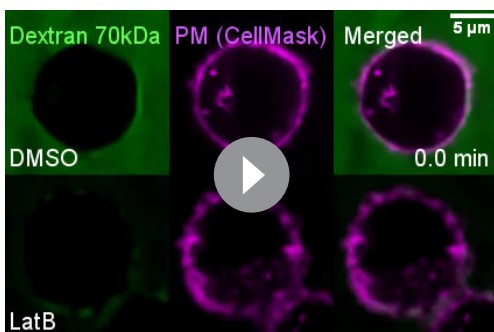

**Video 1.** Live-cell super-resolution structured illumination microscopy (SIM) imaging of CellMask-labeled plasma membrane in suspended MCF-10A cells treated with dimethyl sulfoxide (DMSO) or latrunculin B (LatB) corresponding to *Figure 2G*.
https://elifesciences.org/articles/96178/figures#video1

**Video 2.** Live-cell spinning disk confocal imaging of CellMask-labeled plasma membrane and 70 kDa dextran in suspended MCF-10A cells treated with dimethyl sulfoxide (DMSO) or latrunculin B (LatB) corresponding to *Figure 2H*.
https://elifesciences.org/articles/96178/figures#video2

## Septin and dynamin drive macropinocytosis-like process

We next sought to determine the molecular mechanism underlying GUVac-associated macropinocytosis-like process. The actin cortex, positioned beneath the plasma membrane, has a crucial function in preserving membrane tension. When disrupted, it can result in micron-sized membrane fluctuations in the continuous membrane (*Cannon et al., 2019*; *Weems et al., 2023*). Septins are the only known eukaryotic proteins capable of sensing or inducing micron-scale membrane curvature (*Beber et al., 2019*; *Benoit et al., 2023*; *Cannon et al., 2019*; *Shi et al., 2023*). We thus speculated that a member (or members) of the septin family might play a role in the initial step of macroinocytosis-like process. A recent study demonstrated that fluorescent protein-tagged septin 6 is recruited to the plasma membrane of matrix-detached blebbing cells, specifically to the convex region (from an intracellular viewpoint) located between the blebs, where the plasma membrane exhibits an inward curvature (*Weems et al., 2023*). Using super-resolution SIM analysis at an earlier time point (6 hr) of suspension with LatB, we observed that mCherry-tagged septin 6 was recruited and organized into a semi-circular structure at the inwardly curved plasma membrane. This recruitment was hindered when the cells were treated with forchlorfenuron (FCF), a known septin inhibitor that hinders the assembly and functionality of septin proteins (*Hu et al., 2008*; *Weems et al., 2023*; *Figure 3A and B*). Consistently, treatment with FCF or the co-depletion of two key members of the septin family, septin 2 and septin 9, effectively suppressed GUVac formation in suspended MCF-10A cells exposed to LatB (*Figure 3C and D*). Previous works demonstrated that amphipathic helix (AH) domain enables septins to sense micron-scale membrane curvature (*Cannon et al., 2019*; *Weems et al., 2023*). We found that the deletion mutant of septin 6 lacking the AH domain failed to be recruited to the membrane of LatB-treated suspended MCF-10A cells (*Figure 3E*). These findings suggested that septin 6 (potentially along with other members) is recruited to the fluctuating cell membrane via the AH domain. This recruitment may then lead to the generation of inward plasma membrane curvature at the micron scale and eventual GUVac formation.

Dynamin plays an important role in the formation of endocytic vesicles by pinching them off from the plasma membrane (*Ferguson and De Camilli, 2012*). Given that we observed the pinching off of invaginated portions of the plasma membrane in live super-resolution SIM analysis (*Figure 2G*, *Video 1*, and *Figure 2—figure supplement 1A*), we therefore investigated whether dynamin also contributes to the accumulation of large vacuoles (0.5–1 µm diameter), and thereby GUVac formation in suspended MCF-10A cells treated with LatB. Knockdown of dynamin 2 or forced expression of a dominant negative form of dynamin 2 (DNM2-K44A) resulted in significant inhibition of GUVac formation (*Figure 3F and G*). Furthermore, GUVac formation was consistently suppressed in cells treated with various dynamin inhibitors (Dynasore, Dynole 34-2, OctMAB, and MitMAB) (*Figure 3H*). Together, these results indicated that dynamin-mediated fission of the inwardly curved plasma membrane structures that give rise to the accumulation of multiple vacuoles is required for GUVac formation.

## PI(3)P is required for the fusion of vacuoles underlying GUVac formation

We next investigated the mechanism by which vacuoles fuse and give rise to a single giant vacuole during GUVac formation. Phosphoinositides are a minor component of membrane phospholipids but play a key role in remodeling organelle membranes including membrane fusion (*Posor et al., 2022*). To explore the possible role of phosphoinositides in GUVac formation, we first examined the effects of pharmacological and genetic manipulation of PI 3-kinase (PI3K). Treatment with the broad-spectrum PI3K inhibitor LY294002 or the class III PI3K inhibitor VPS34-IN1 resulted in significant attenuation of GUVac formation in LatB-treated suspended MCF-10A cells, whereas the class I PI3K-specific inhibitors GDC-0941 (an inhibitor of p110) or TGX-221 (an inhibitor of p110β) did not (*Figure 4A*). We further revealed that knockdown of class II PI3Kα (PI3K-C2α) or class III PI3K (VPS34), but not that of class I PI3K subunits (p110α or p110β), resulted in marked inhibition of GUVac formation (*Figure 4B*, *Figure 4—figure supplement 1A–C*). Moreover, CRISPR/Cas9-mediated knockout (KO) of PI3K-C2α or VPS34 in MCF-10A cells greatly attenuated GUVac formation (*Figure 4C*, *Figure 4—figure supplement 1D and E*).

Phosphatidylinositol 3-phosphate [PI(3)P] can be generated on endocytic vesicles by either VPS34 or PI3K-C2α (*Boukhalfa et al., 2020*; *Campa et al., 2018*; *Posor et al., 2022*) and plays a key role in the fusion of these vesicles (*Orr and Wickner, 2023*; *Posor et al., 2022*). We therefore reasoned that

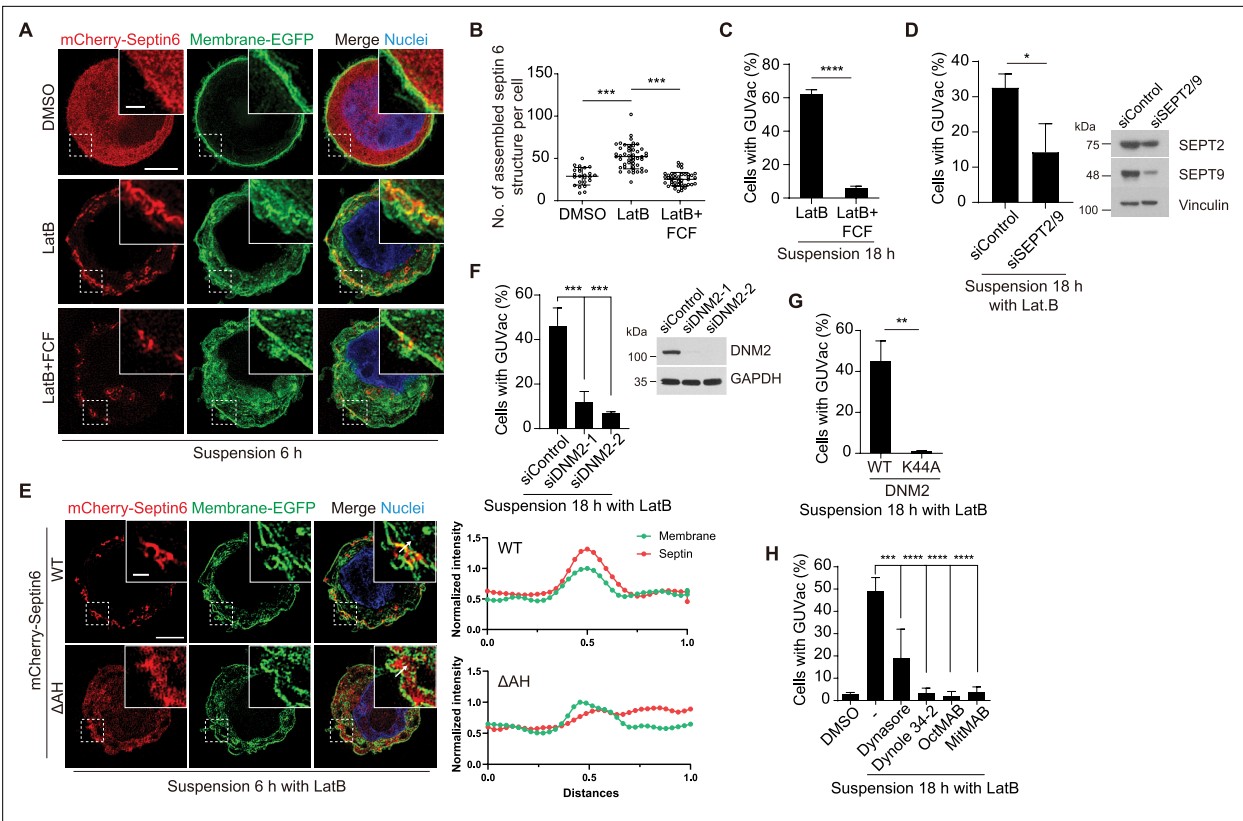

**Figure 3.** Recruitment of septin to the fluctuating plasma membrane drives macropinocytosis-like phenomenon. (**A**) Representative super-resolution structured illumination microscopy (SIM) fluorescence images (maximum intensity projection) of MCF-10A cells expressing membrane-targeted EGFP and mCherry–septin 6 treated with indicated drugs for 6 hr. Scale bars, 5 μm (main panels) or 1 μm (inset). (**B**) Number of assembled mCherry-septin 6 structures (filaments and collar-like) per cell, quantified in experiments as in (**A**). Dimethyl sulfoxide (DMSO) (n = 24), latrunculin B (LatB) (n = 47), LatB + forchlorfenuron (FCF) (n = 41). Each data point represents an individual cell analyzed from two independent experiments. (**C**) Percentage of suspended MCF-10A cells showing giant unilocular vacuole (GUVac) formation after incubation with the indicated drugs for 18 hr. LatB (n = 783), LatB + FCF (n = 1016). (**D**) Percentage of GUVac formation in 18 hr suspended MCF-10A cells after co-depletion of Septin 2 and 9 or control knockdown. siControl (n = 1270) and siSEPT2/9 (n = 511). (**E**) Representative super-resolution SIM fluorescence images (maximum intensity projection) of MCF-10A cells expressing membrane-targeted EGFP and either mCherry-septin 6 WT or a mutant lacking the AH domain (ΔAH), which were suspended in the presence of LatB for 6 hr. Scale bars, 5 μm (main panels) or 1 μm (inset). The profiles on the right side represent the normalized intensity of each fluorescence signal along the indicated arrow from the inset of the merge panel. (**F**) Percentage of cells showing GUVac formation for MCF-10A cells that had been transfected with control or dynamin 2 siRNAs and treated with LatB in suspension culture for 18 hr (left panel). siControl (n = 746), siDNM2-1 (n = 822), and siDNM2-2 (n = 712). Immunoblot analysis of dynamin 2 in MCF-10A cells transfected with control or two independent dynamin 2 siRNAs (right panel). (**G**) Percentage of cells showing GUVac formation for MCF-10A cells expressing WT or dominant negative mutant (K44A) forms of dynamin 2 that had been suspended in the presence of LatB for 18. WT (n = 607) and K44A (n = 737). (**H**) Percentage of MCF-10A cells showing GUVac formation after suspension in the presence of the indicated inhibitors and incubation for 18 h. DMSO (n = 1423), LatB (n = 1151), LatB + Dynasore (n = 1407), LatB + Dynole 34-2 (n = 947), LatB + OctMAB (n = 699), and LatB + MitMAB (n = 1404). All quantitative data are means ± SD. The n values represent the total number of cells examined in two (**B**) or three independent experiments (**C, D, F–H**). **p<0.01, ***p<0.001, ****p<0.0001 by one-way ANOVA with Tukey's multiple comparisons (**B, F, H**) or two-tailed unpaired *t*-test (**C, D, G**).

The online version of this article includes the following source data for figure 3:

**Source data 1.** Quantification data corresponding to *Figure 3B*.

**Source data 2.** Quantification data corresponding to *Figure 3C*.

**Source data 3.** Quantification data corresponding to *Figure 3D*.

**Source data 4.** Quantification data corresponding to *Figure 3F*.

**Source data 5.** Quantification data corresponding to *Figure 3G*.

**Source data 6.** Quantification data corresponding to *Figure 3H*.

**Source data 7.** Uncropped blot images with sample labeling used in *Figure 3D*.

**Source data 8.** Original blot images used in *Figure 3D*.

*Figure 3 continued on next page*

*Figure 3 continued*

**Source data 9.** Uncropped blot images with sample labeling used in *Figure 3F*.

**Source data 10.** Original blot images used in *Figure 3F*.

the generation of PI(3)P by these enzymes might contribute to vacuole fusion during GUVac formation. To investigate this possibility, we applied TEM and super-resolution SIM to observe the size of vacuoles in suspended VPS34 KO, PI3K-C2α KO, and VPS34-IN1-treated cells at various time points after LatB treatment. We detected the formation of internal vacuoles at 1, 3, and 6 hr in these KO and VPS34-IN1-treated cells as in the WT and vehicle-treated cells. However, in contrast to WT and vehicle-treated cells, in which the vacuoles fused to form a single giant vacuole (i.e., GUVac) over time, the vacuoles in the KO and VPS34-IN1-treated cells did not increase in size even after 12–24 hr of LatB treatment, resulting in the failure of GUVac formation (*Figure 4D–F*). Finally, the vacuoles induced in suspended cells by LatB treatment were positive for PI(3)P but not for $PI(3,4)P_2$ (which can also be generated by PI3K-C2α), as revealed by expression of the specific probes mCherry-2xFYVE and mCherry–PH-TAPP1 (*Posor et al., 2022*), respectively (*Figure 4G*). Collectively, these data indicated that PI(3)P produced on the surface of vacuoles by VPS34 and PI3K-C2α seems to be necessary for the vacuole fusion step during GUVac formation.

## GUVac formation confers anoikis resistance

We next evaluated the physiological relevance of GUVac formation under the stress imposed by changes in matrix attachment and F-actin architecture. Vacuolization in mammalian cells has been suggested to be associated with cytotoxicity and cell death (*Johnson et al., 2011*; *Kar et al., 2009*; *Papini et al., 1994*; *Shubin et al., 2016*). We therefore first tested whether cells with GUVacs are undergoing cell death by immunostaining for cleaved caspase-3, a marker of apoptotic death. Suspended MCF-10A cells under the control condition were typically aggregated and negative for cleaved caspase-3 (*Figure 5A–C*), consistent with previous data showing that cadherin-mediated cell–cell adherens junctions prevent anoikis (*Hofmann et al., 2007*). In contrast, treatment of suspended cells with EDTA, a $Ca^{2+}$ chelator that blocks cadherin function, resulted in significant suppression of cell aggregation (without induction of GUVac formation) and thereby triggered anoikis associated with cleavage of caspase-3. Treatment of suspended cells with LatB also prevented cell aggregation, similar to the effect of EDTA, but it did not induce cleavage of caspase-3 (*Figure 5A–C*). We also detected activation of survival signaling by the kinase AKT and the antiapoptotic protein BCL2 in the LatB-treated cells (*Figure 5—figure supplement 1*). To substantiate the finding that matrix-detached cells with the ability to form GUVacs can avoid cell death, we further measured cell viability with a trypan blue exclusion assay. We obtained similar results showing that cells with LatB-induced GUVac formation were protected from EDTA-induced cell death even after prolonged suspension culture (*Figure 5D–F*). Treatment with soluble Fas ligand (sFasL) can induce cell death by the extrinsic apoptotic pathway (*Kischkel et al., 1995*), and we therefore also tested whether GUVac formation allows cells to escape such ligand-induced cell death. Simultaneous treatment of matrix-deprived MCF-10A cells with both LatB and sFasL for 24 hr did not prevent sFasL-induced cell death or result in GUVac formation (*Figure 5G*). In contrast, prior treatment of the cells with LatB for 24 hr (a condition that allowed GUVac formation) prevented cell death induced by subsequent treatment with sFasL (*Figure 5G*), suggesting that GUVac formation suppresses cell death signaling. Furthermore, PI3K-C2α KO and VPS34 KO cell lines, which are defective in GUVac formation (*Figure 4C*), were susceptible to EDTA-induced cell death even in the presence of LatB (*Figure 5H and I*), confirming that GUVac formation promotes cell survival under stress associated with altered matrix and actin environments. We also monitored the fate of cells with GUVacs after they were returned to the normal adherent condition in the absence of LatB. Time-lapse live imaging revealed that cell adherence to the matrix was followed by the gradual breakup of the GUVac into smaller vacuoles, which were in turn eventually eliminated in association with the adoption of a normal adherent morphology (*Figure 5J*). These observations indicated that GUVac formation is a reversible process that transiently protects cells from a survival-threatening situation.

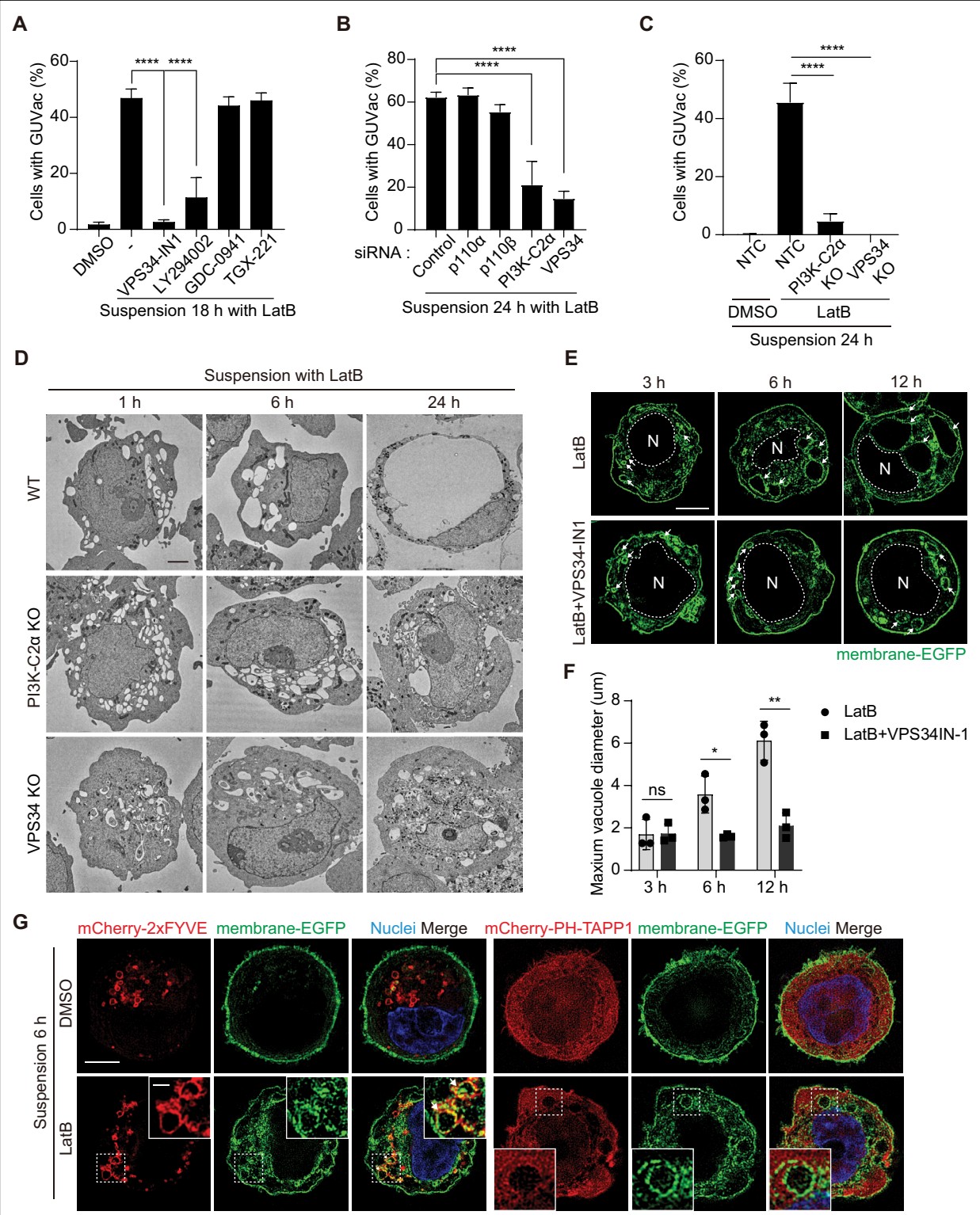

**Figure 4.** PI(3)P is required for vacuole fusion during giant unilocular vacuole (GUVac) formation. (**A**) Percentage of suspended MCF-10A cells showing GUVac formation after treatment with the indicated drugs for 18 hr. Dimethyl sulfoxide (DMSO) (n = 1122), latrunculin B (LatB) (n = 1386), LatB + VPS34-IN1 (n = 989), LatB + LY294002 (n = 876), LatB + GDC-0941 (n = 787), and LatB + TGX-221 (n = 801). (**B**) Percentage of cells showing GUVac formation for MCF-10A cells transfected with the indicated siRNAs and suspended in the presence of LatB for 24 hr. siRNAs: control (n = 456), p110α (n = 718), p110β (n = 709), PI3K-C2α (n = 421), and VPS34 (n = 295). (**C**) Percentage of cells showing GUVac formation for nontargeting control (NTC), PI3K-C2α KO, or VPS34 KO MCF-10A cells suspended in the presence of DMSO or LatB for 24 hr. NTC/DMSO (n = 746), NTC/LatB (n = 730), PI3K-C2α KO/LatB

*Figure 4 continued on next page*

*Figure 4 continued*

(n = 824), and VPS34 KO/LatB (n = 939). (**D**) Representative transmission electron microscopy (TEM) images of suspended WT, PI3K-C2α KO, or VPS34 KO MCF-10A cells treated with LatB for the indicated times. Scale bars, 2 µm. (**E**) Representative super-resolution structured illumination microscopy (SIM) fluorescence images of MCF-10A cells expressing membrane-targeted EGFP that had been suspended in the presence of the indicated drugs for the indicated times. The nucleus is denoted by the letter 'N' and outlined with a dashed line. Arrows indicate vacuoles. Scale bars, 5 µm. (**F**) Maximum diameter of vacuoles in each cell for experiments as in (**E**). LatB: 3 hr (n = 29), 6 hr (n = 39), and 12 hr (n = 36). LatB + VPS34-IN1: 3 hr (n = 47), 6 hr (n = 43), and 12 hr (n = 39). (**G**) Representative super-resolution SIM fluorescence images of MCF-10A cells expressing membrane-targeted EGFP and either mCherry-2xFYVE or mCherry–PH-TAPP1 that had been suspended in the presence of DMSO or LatB for 6 hr. All quantitative data are means ± SD. The *n* values represent the total number of cells examined in three independent experiments (**A–C**). *p<0.05, **p<0.01, ****p<0.0001; ns, not significant by one-way ANOVA with Tukey's multiple comparisons (**A–C**) or two-tailed unpaired *t*-test (**F**).

The online version of this article includes the following source data and figure supplement(s) for figure 4:

**Source data 1.** Quantification data corresponding to *Figure 4A*.

**Source data 2.** Quantification data corresponding to *Figure 4B*.

**Source data 3.** Quantification data corresponding to *Figure 4F*.

**Figure supplement 1.** Verification of PI3K knockdown or knockout in MCF-10A cell.

**Figure supplement 1—source data 1.** Uncropped blot images with sample labeling used in *Figure 4—figure supplement 1C*.

**Figure supplement 1—source data 2.** Original blot images used in *Figure 4—figure supplement 1C*.

**Figure supplement 1—source data 3.** Uncropped blot images with sample labeling used in *Figure 4—figure supplement 1D*.

**Figure supplement 1—source data 4.** Original blot images used in *Figure 4—figure supplement 1D*.

**Figure supplement 1—source data 5.** Uncropped blot images with sample labeling used in *Figure 4—figure supplement 1E*.

**Figure supplement 1—source data 6.** Original blot images used in *Figure 4—figure supplement 1E*.

## Discussion

The mechanisms by which cells acquire resistance to anoikis remain poorly understood. We have now demonstrated the emergence of a hitherto unrecognized organelle, namely GUVac, in mammary epithelial cells, as well as in other cell lines from the secretory organs, triggered by the detachment from the ECM. GUVac formation is initiated by an unconventional macropinocytosis-like process that requires actin depolymerization and septin recruitment to the plasma membrane to achieve micronsized inward membrane curvature. The resulting vacuoles that accumulate in the cell gradually merge in a manner dependent on the activity of VPS34 and PI3K-C2α, eventually forming a single giant membrane-bound sac that occupies a large proportion of the cell volume. Importantly, cells that have developed a GUVac in response to the loss of attachment and disruption of actin architecture manifest resistance to anoikis (*Figure 5—figure supplement 2*). Furthermore, the formation of GUVac is reversible when cells return to normal adherent conditions, suggesting that GUVac formation to support the survival of circulating tumor cells (CTCs) might be reversed when the cells seed the metastatic niche.

CTCs are cancer cells that have been shed from a primary tumor and circulate in the bloodstream. Sustained detachment from ECM and hemodynamic shear stress threaten the survival of CTCs, with the result that only a small proportion of these cells escape cell death during circulation in the blood (*Massagué and Obenauf, 2016*). A previous study of the effects of fluid shear stress on the viability of CTCs revealed that surviving cells manifest diminished F-actin assembly (*Xin et al., 2019*). Given our current finding that the inhibition of F-actin assembly in suspended cells promotes GUVac formation, it is plausible that surviving CTCs induce GUVac formation to maintain their viability in the face of hemodynamic shear stress in the blood. Clinical observations have provided evidence of large vacuoles in some breast CTCs (*Kolostova et al., 2016*). Furthermore, recent research has shown that intracellular microbiota of circulating breast cancer cells promotes host cell survival by increasing resistance to fluid shear stress through disruption of the actin cytoskeleton, thereby facilitating metastatic colonization in breast cancer (*Fu et al., 2022*). Given the ability of various bacterial toxins to disrupt the actin cytoskeleton of host cells (*Barbieri et al., 2002*; *Harterink et al., 2017*; e.g., SpvB shown in *Figure 2F and G*), it will be of interest to explore whether breast CTCs harboring intracellular bacteria are able to form GUVacs during their circulation in the bloodstream and whether such potential GUVac formation is reversible after the cells have seeded the metastatic niche. CTCs are also eliminated by immune cells and require strategies to escape from this immune surveillance if they are to establish metastases (*Mohme et al., 2017*). The interaction of FasL expressed on natural killer cells and cytotoxic T cells

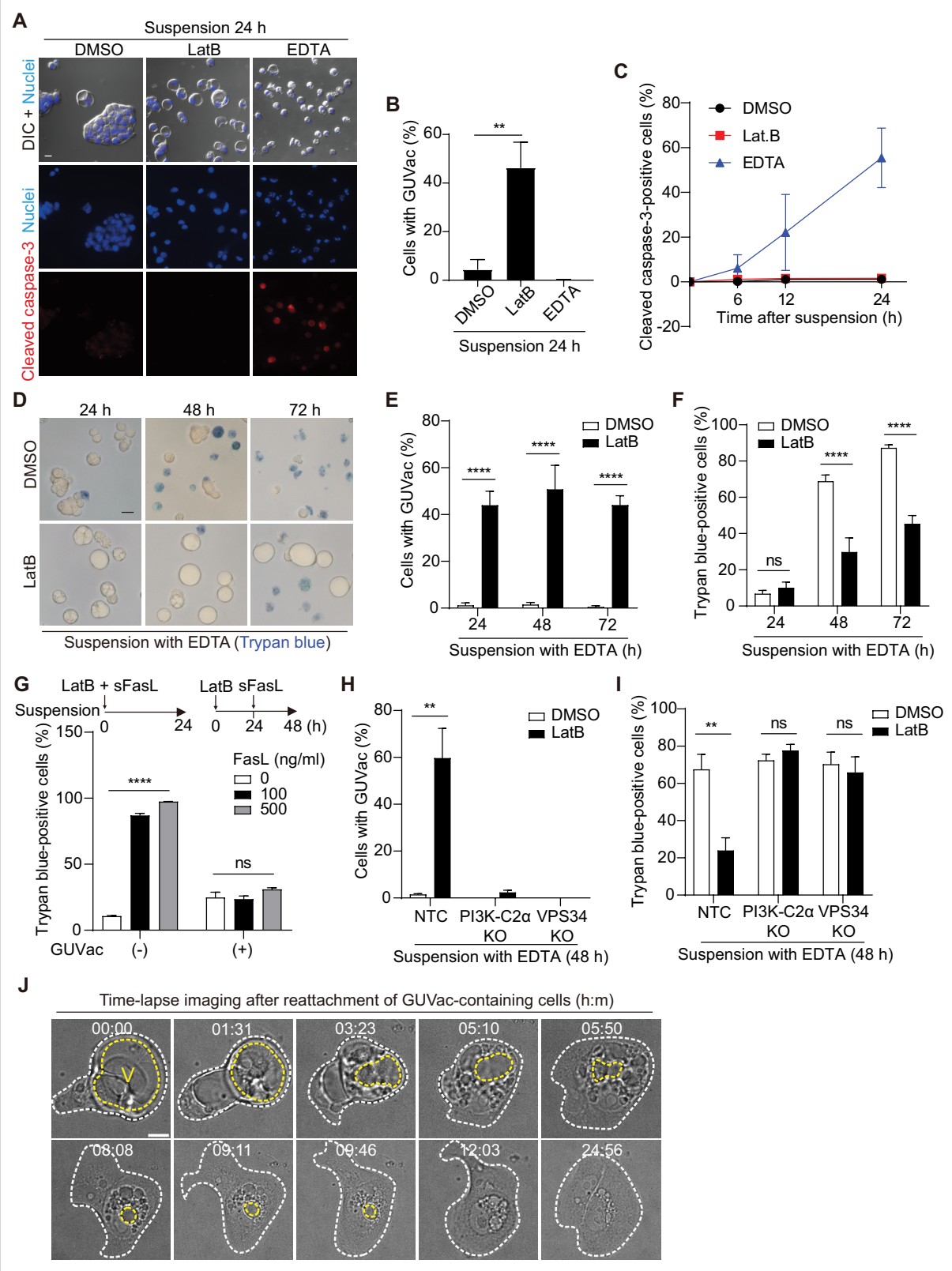

**Figure 5.** Giant unilocular vacuole (GUVac) formation promotes cell survival in altered actin and matrix environments. (**A–C**) MCF-10A cells were suspended in the presence of latrunculin B (LatB) or EDTA for the indicated times and then immunostained for cleaved caspase-3. Nuclei were stained with Hoechst 33342. Representative differential interference contrast (DIC) and fluorescence images at 24 hr (scale bar, 20 μm) (**A**), the percentage of cells positive for GUVac formation at 24 hr (**B**), and the time course for the percentage of cells positive for cleaved caspase-3 (**C**) are shown. Dimethyl

*Figure 5 continued on next page*

*Figure 5 continued*

sulfoxide (DMSO): 0 hr (n = 1030), 6 hr (n = 1742), 12 hr (n = 1423), and 24 hr (n = 1273). LatB: 6 hr (n = 1479), 12 hr (n = 1508), and 24 hr (n = 1443). EDTA: 6 hr (n = 1976), 12 hr (n = 1578), and 24 hr (n = 1296). (**D–F**) MCF-10A cells were suspended with EDTA and in the presence of DMSO or LatB for the indicated times and then stained with trypan blue. Representative bright-field images (scale bar, 20 µm) (**D**) as well as the percentage of cells positive for GUVac formation (**E**) and the percentage of trypan blue-positive cells (**F**) are shown. DMSO: 24 hr (n = 1003), 48 hr (n = 797), and 72 hr (n = 943). LatB: 24 hr (n = 1473), 48 hr (n = 2100), and 72 hr (n = 1350). (**G**) Percentage of trypan blue-positive cells for suspended MCF-10A cells that either were simultaneously treated with both LatB and the indicated concentrations of sFasL for 24 hr (a condition under which cell death signaling precedes GUVac formation) or were treated with LatB for 24 hr and then incubated in the presence of sFasL for 24 hr (a condition under which GUVac formation precedes cell death signaling). GUVac(–): sFasL at 0 ng/ml (n = 1193), 100 ng/ml (n = 1441), or 500 ng/ml (n = 994). GUVac(+): sFasL at 0 ng/ml (n = 1240), 100 ng/ml (n = 1418), or 500 ng/ml (n = 1529). (**H, I**) Nontargeting control (NTC), PI3K-C2α KO, and VPS34 KO MCF-10A cells were suspended with EDTA in the presence of DMSO or LatB for 48 hr, after which the percentage of cells showing GUVac formation (**H**) and the percentage of trypan blue-positive cells (**I**) were determined. NTC: DMSO (n = 927), LatB (n = 1369). PI3K-C2α KO: DMSO (n = 627), LatB (n = 883). VPS34 KO: DMSO (n = 1026), and LatB (n = 885). (**J**) Time-lapse bright-field images of cells with GUVacs after matrix reattachment. MCF-10A cells were suspended in the presence of LatB for 24 hr, washed to eliminate LatB, and then cultured in an adhesive confocal dish for 14 hr before imaging for the indicated times (hour:minute). The yellow dotted line represents the largest vacuole in each cell, while the white dotted line indicates the spread cell area. Scale bar, 10 µm. All quantitative data are means ± SD. The *n* values represent the total number of cells examined in three (**B**, **C**, **G–I**) or four (**E**, **F**) independent experiments. **$p<0.01$, ****$p<0.0001$; ns, not significant (two-tailed unpaired *t*-test). V, vacuole.

The online version of this article includes the following source data and figure supplement(s) for figure 5:

**Source data 1.** Quantification data corresponding to *Figure 5B and C*.

**Source data 2.** Quantification data corresponding to *Figure 5E and F*.

**Source data 3.** Quantification data corresponding to *Figure 5G*.

**Source data 4.** Quantification data corresponding to *Figure 5H and I*.

**Figure supplement 1.** Immunoblot analysis of survival signaling in MCF-10A cells suspended with EDTA and in the presence of dimethyl sulfoxide (DMSO) or latrunculin B (LatB) for the indicated times.

**Figure supplement 1—source data 1.** Uncropped blot images with sample labeling used in *Figure 5—figure supplement 1*.

**Figure supplement 1—source data 2.** Original blot images used in *Figure 5—figure supplement 1*.

**Figure supplement 2.** Hypothetical model for the formation of giant unilocular vacuole (GUVac) that enhances the anoikis resistance.

with Fas on CTCs triggers apoptosis in the latter cells (*Pereira-Veiga et al., 2022*). Of interest in this regard, we found that treatment with sFasL failed to induce apoptosis in mammary epithelial cells containing GUVacs, suggesting that GUVac formation might also function as a mechanism of immune evasion in CTCs. Whether GUVacs are indeed present in CTCs of breast cancer patients and whether such GUVac formation is associated with metastasis warrant further investigation.

## Materials and methods
### Plasmids

Complementary DNAs encoding mCherry-H2B and membrane-targeted EGFP tagged with Igκ leader sequence at the N-terminal and transmembrane domain of platelet-derived growth factor receptor at the C-terminal were cloned into the pBABE-puro vector. A cDNA for GFP-tagged mouse septin 6 was obtained from Addgene (plasmid #38296) and cloned into the pLVX-IRES-puro vector (Clontech); the GFP sequence was then replaced with the mCherry sequence to generate a construct encoding mCherry–septin 6. pLVX-IRES-puro-mCherry-ΔAH septin 6 was generated by removing amino acids 355–372 from pLVX-IRES-puro-mCherry-septin 6 through PCR. The cDNAs for WT and dominant negative (K44A) forms of rat dynamin 2 were cloned into the pMSCV-puro vector. The cDNAs for GFP-tagged two FYVE domains of mouse HRS and PH domain of human TAPP1 were obtained from Addgene (#140047 and #161985, respectively) and cloned into the pLVX-IRES-puro vector; the GFP sequence was then replaced with mCherry sequence to generate constructs encoding mCherry-2xFYVE and mCherry-PH-TAPP1. The cDNAs for eGFP-tagged SpvB (*Salmonella* SpvB 375–591) sequence were obtained from Addgene (#89446) and cloned into the pMSCV-puro vector.

### Cell culture

MCF-10A cells were cultured in Dulbecco's modified Eagle's medium (DMEM)-F12 (Welgene, LM002-04) supplemented with 5% horse serum (Invitrogen, 26050-088), epidermal growth factor (Peprotech,

AF-100-15) at 20 ng/ml, insulin (Gibco, A11382II) at 10 µg/ml, hydrocortisone (Sigma, H0888) at 0.5 µg/ml, and cholera toxin (List Biological Laboratories, 100B) at 100 ng/ml. HEK293T, MDCK, HFE-145 (gastric epithelial) cells and PANC-1 (pancreatic adenocarcinoma) cells were cultured in DMEM medium supplemented with 10% fetal bovine serum (FBS) (Gibco, 12483020). AGS, MKN-45 (both from gastric adenocarcinoma), and BCPAP (thyroid carcinoma) were cultured in RPMI medium supplemented with 10% FBS. HMEpiCs (PCS-600-010) isolated from the normal breast tissue of adult female were cultured in Mammary Epithelial Cell Basal Medium (ATCC, PCS-600-030) supplemented with Mammary Epithelial Cell Growth Kit (ATCC, PCS-600-040). RPE-1 cells were cultured in DMEM-F12 supplemented with 10% FBS. Human umbilical vein endothelial cells (HUVECs) were cultured in EGM-2MV (Lonza, CC-3202). BJ-hTERT and IMR-90 cells were cultured in DMEM supplemented with 10% FBS and 2 mM L-glutamine (Gibco, 25030081). All media were supplemented with penicillin (100 U/ml)–streptomycin (100 µg/ml) (Gibco, 15140122). HFE-145 cells were kindly provided by Dr. Hassan Ashktorab. AGS and MKN-45 cell lines were purchased from a Korean cell line bank, and all other cell lines, including HMEpiCs and HUVECs, were obtained from the American Type Culture Collection. Cell line authentication through STR profiling was performed by the respective providers. Cells used in this study were maintained under 20 passages before utilization and regularly tested for mycoplasma contamination by MycoStrip mycoplasma detection kit (Invitrogen, rep-mys).

## Viral infection

For retrovirus production, VSV-G, Gag-Pol, pBABE-puro, or pMSCV-puro-based constructs were transiently transfected in HEK293T cells using polyethylenimine (Polysciences, 24765). For lentivirus production, pMD2.G, psPAX2, and pLVX-IRES-puro-based constructs were employed. Viral supernatant was collected and added to MCF-10A cells with polybrene (Merck, H9268) at 10 µg/ml.

## Suspension cell culture

Cells ($4 \times 10^4$) were isolated by exposure to trypsin-EDTA (Gibco, 15400054), dissociated from each other, and seeded in six-well Cell Floater plates (SPL Life Sciences, 39706). For DIC and fluorescence microscopy analysis, suspended cells collected at the indicated time points were isolated by centrifugation and rapidly reseeded on poly-L-lysine (Sigma, P4832)-coated coverslips for 3 min before fixation.

## Reagents

Reagents were obtained from the following sources and used at the indicated concentrations: LatB (Sigma, 5288, used at 5 µM), cytochalasin D (Sigma, C8273, used at 0.5 µM), jasplakinolide (Tocris, 2792, used at 100 nM), Rho inhibitor I (Cytoskeleton, CT04-A, used at 2 µg/ml), Y-27632 (Sigma, Y0503, used at 20 µM), EHT 1864 (Tocris, 3872, used at 2 µM), ML 141 (Tocris, 4266, used at 10 µM), blebbistatin (Sigma, B0506, used at 50 µM), nocodazole (Sigma, M1404, used at 1 µg/ml), EIPA (Sigma, A3085, used at 50 µM), FITC–dextran (70 kDa) (Sigma, 46945, used at 1 mg/ml), tetramethylrhodamine-dextran (70 kDa) (Invitrogen, D1818, used at 0.1 mg/ml), LysoTracker Red (Invitrogen, L7528, used at 50 nM), CellMask Plasma Membrane Stains Deep Red (Invitrogen, C10046), LY294002 (Sigma, L9908, used at 50 µM), VPS34-IN1 (Cayman, 17392, used at 5 µM), TGX-221 (Cayman, 10007349, used at 50 nM), GDC-0941 (Selleckchem, S1065, used at 100 nM), Dynasore (Sigma, D7693, used at 100 µM), Dynole 34-2 (Tocris, 4222, used at 10 µM), OctMAB (Tocris, 4225, used at 20 µM), MitMAB (Abcam, ab120466, used at 10 µM), FCF (Sigma, C2791, used at 100 µM), Matrigel (Corning, 354230), and recombinant human sFasL (Peprotech, 310-03H).

## Antibodies

Antibodies to BCL2 (15,071T), Bim (2933T), phospho-Akt (Ser473, 4060S; Thr308, 13038S), cleaved caspase-3 (9664S), Na,K-ATPase 1 (3010S), Vinculin (13901S), and VPS34 (4263S) were obtained from Cell Signaling Technology; those to CHC (ab21679), LAMP1 (ab25630), LC3B (ab192890), glyceraldehyde-3-phosphate dehydrogenase (GAPDH) (ab8245), and GFP (ab13970) were from Abcam; those to dynamin 2 (sc-166669) and PI3K-C2α (sc-365290) were from Santa Cruz Biotechnology; those to EEA1 (610457) were from BD Biosciences; those to Transferrin receptor (13-6800) were from Invitrogen; and those to β-actin (A5316) were from Sigma-Aldrich.

## Trypan blue exclusion assay

Suspended cells were collected at the indicated time points, isolated by centrifugation, suspended in 10 µl of culture medium, and then stained with 10 µl of trypan blue (0.4%, Invitrogen, T10282). The stained cells were plated in cell counting chamber slides (Thermo Fisher, C10283) and examined by light microscopy.

## siRNA transfection

Cells were transfected for 2 days with siRNAs (synthesized by GenePharma) at 20 nM with the use of the Lipofectamine RNAiMAX reagent (Invitrogen, 13778150) and were then suspended. The siRNA sequences were as follows: control, 5'-CGUACGCGGAAUACUUCGA-3'; p110α, 5'-GCCAGUACCUCA UGGAUUA-3'; p110β, 5'-GAAUCCAAUGGGAACUGUU-3'; PI3K-C2α, 5'-CAUCUACAGAACCUAU AUA-3'; VPS34, 5'-GGAUAUCAACGUCCAGCUU-3'; endophilin A2, 5'-CAUGCUCAACACGGUG UCCAA-3' (siEndoA2-2) or 5'-UACACUAGCGCUGACUCCCAA-3' (siEndoA2-3); dynamin 2, 5'-GGCU GACCAUCAACAACAU-3' (siDNM2-1) or 5'-GCAGCCAGAAGGAUAUUGA-3' (siDNM2-2); CHC, 5'-GCUGGGAAAACUCUUCAGA-3' (siCHC_1) or 5'-UAAAUUUCCGGGCAAAGAG-3' (siCHC_2); Septin2, 5'-GAGGCUUCAACUGUUGAAAUU-3'; and Septin9, 5'-GCACGAUAUUGAGGAGAAAUU-3'.

## CRISPR/Cas9-mediated genome engineering

Single-guide RNA oligonucleotides were cloned into the pSpCas9-2A-GFP vector (Addgene #48138). The guide sequences were as follows: NTC (nontargeting control), 5'-GCGGAGCTAGAGAGCGGTCA -3'; PI3K-C2α, 5'-ACCCGATACGAACATGTTTT-3'; and VPS34, 5'-TTCCCTAGCATGTTTCGCCA-3'. MCF-10A cells were transfected with the plasmids with the use of ViaFect (Promega, E4982). GFP-expressing cells were sorted into single-cell clones by flow cytometry with a Moflo Astrios EQ instrument (Beckman Coulter). Clones were expanded and assessed for gene knockout by immunoblot analysis.

## Immunoblot analysis

Cells were lysed with RIPA buffer (50 mM Tris-Cl [pH 7.5], 150 mM NaCl, 1 mM EDTA, 0.1% SDS, 1% Triton X-100, 0.5% sodium deoxycholate, 10 mM NaF, 1 mM $Na_3VO_4$) supplemented with protease inhibitors (1 mM phenylmethylsulfonyl fluoride, leupeptin [1 µg/ml], pepstatin [1 µg/ml]). Cell lysates were centrifuged at $16,000 \times g$ for 10 min at 4°C, and the resulting supernatants were collected and assayed for protein concentration with a BCA Protein Assay Kit (Thermo Fisher, 23225). Samples containing equal amounts of protein were denatured by boiling for 10 min with Laemmli sample buffer and then fractionated by SDS-polyacrylamide gel electrophoresis, and the separated proteins were transferred to a polyvinylidene difluoride membrane. The membrane was exposed for 1 hr at room temperature to 5% skim milk in phosphate-buffered saline containing 0.1% Tween-20 (PBST) before incubation overnight at 4°C with primary antibodies. The membrane was washed three times with PBST and then incubated for 1 hr at room temperature with Peroxidase AffiniPure Goat Anti-Rabbit IgG (H+L) (Jackson ImmunoResearch Laboratories, 113-035-003) or Goat Anti-Mouse IgG (H+L) Secondary Antibody, HRP (Invitrogen, 31430). Immune complexes were detected with Immobilon Forte Western HRP Substrate (Millipore, WBLUF0500).

## RNA isolation and quantitative RT-PCR analysis

Total RNA was isolated from cells with the use of RiboEx (GeneAll, 301-902), and 1 µg of the purified RNA was subjected to reverse transcription (RT) with the use of a TOPscript cDNA synthesis kit (Enzynomics, EZ005M). The resulting cDNA was subjected to real-time PCR analysis with TOPreal SYBR Green qPCR preMIX (Enzynomics, RT500M) and a Bio-Rad CFX Connect system. The PCR primers (forward and reverse, respectively) for human genes were as follows: GAPDH, 5'-CTTCGCTC TCTGCTCCTCCT-3' and 5'-GTTAAAAGCAGCCCTGGTGA-3'; p110α, 5'-AGTAGGCAACCGTGAA GAAAAG-3' and 5'-GAGGTGAATTGAGGTCCCTAAGA-3'; and p110β, 5'-TATTTGGACTTTGCGA CAAGACT-3' and 5'-TCGAACGTACTGGTCTGGATAG-3'. The amount of p110α or p110β mRNA was normalized by that of GAPDH mRNA.

## Immunofluorescence analysis

Cells were seeded on coverslips, fixed with 4% paraformaldehyde for 15 min at room temperature. LysoTracker was pre-stained for 2 hr at 37°C before fixation. Cells were exposed for 1 hr at room

temperature to PBS containing 5% donkey serum, 1% bovine serum albumin, and 0.05% saponin before incubation overnight at 4°C with primary antibodies. For EEA1 staining, cells were blocked with PBS containing 3% bovine serum albumin, and 0.3% Triton X-100. For LC3B staining, cells were fixed with cold methanol and then blocked with PBS containing 3% BSA. The cells were washed three times with PBS containing 0.05% saponin, 0.3% Triton X-100, or just PBS and incubated for 1 hr at room temperature with secondary antibodies (donkey anti-mouse Alexa Fluor 647 [Invitrogen, A31571], donkey anti-mouse Alexa Fluor 594 [Invitrogen, A21203], donkey anti-rabbit Alexa Fluor 594 [Invitrogen, A21207], or donkey anti-chicken Alexa Fluor 488 [Thermo Fisher, A17584]) or Alexa Fluor 594-labeled phalloidin (Invitrogen, A12381), washed another three times, stained with Hoechst 33342 (Invitrogen, H3570), and mounted on glass slides with Prolong Gold antifade reagent (Invitrogen, P36930) for observation with a Nikon Eclipse Ti2 microscope. To quantitatively determine the number of assembled septin 6 structures upon LatB treatment in suspended cells, images were blurred with a Gaussian filter (sigma = 2), background was subtracted with a rolling ball radius of 50 pixels, and segmented using the Find Maxima tool with prominence = 20 in Fiji.

## Super-resolution 3D-SIM imaging

Cells were seeded on 35 mm confocal dishes and processed for immunofluorescence analysis as described above. The 3D-SIM images were obtained using a Ti-2 inverted microscope (Nikon) equipped with the Ziva Light Engine (Lumencor) as a light source. The Ziva Light Engine generates 405, 488, 561, and 640 nm lasers for excitation. To verify the alignment of multiple channels, we utilized multi-spectrum microbeads (TetraSpeck Microspheres, 0.1 μm, T7279) before imaging. The raw images consisted of 15 individual frames, which were captured under five distinct patterned illuminations and three different illumination angles. Sequential image acquisition was performed based on the excitation channel using a high-resolution sCMOS camera with a large field of view (Hamamatsu Photonics K.K., ORCA-Flash4.0 sCMOS camera, 2048 × 2048 pixels). Subsequently, the acquired 3D-SIM images were reconstructed using NIS-Elements (Nikon) software.

## Live-cell imaging by time-lapse microscopy

MCF-10A cells stably expressing mCherry-H2B and membrane-targeted EGFP were seeded with LatB in 35 mm confocal dishes (SPL Life Sciences, 210350) and imaged with a spinning-disk confocal microscope (Nikon CSU-W1) equipped with a live-cell environmental chamber (Okolab) containing a humidified atmosphere of 5% $CO_2$ and maintained at 37°C. Images were acquired every 10 min for 24 hr and were processed with ImageJ software. For live-cell imaging of the reattachment of GUVac-containing cells, the cells were washed several times with fresh culture medium to remove LatB and reseeded on poly-L-lysine-coated 35 mm confocal dishes. After 14 hr, adherent GUVac-containing cells were subjected to confocal imaging as described above. Images were acquired every 5 min for 24 hr and were processed with ImageJ software. For live-cell imaging of CellMask-labeled plasma membrane and 70 kDa tetramethylrodamine (TMR)-dextran, MCF-10A cells were trypsinized, collected, and resuspended with media containing CellMask Plasma Membrane Stains Deep Red (1:1,000). To acquire time-lapse images of live suspended cells in fixed positions right after treatment with DMSO or LatB, cells were suspended on confocal dishes coated in fibronectin. The dishes were incubated at 37°C for 5 min, which allowed cells to attach to the dishes with the round cell shape slightly. The media were changed to new ones containing CellMask (1:10,000), 70 kDa TMR-dextran (0.1 mg/ml), and DMSO or LatB, and the cells were immediately imaged with a spinning-disk confocal or SIM microscope. Images were acquired every 30 s for 1 hr. The images obtained from a spinning-disk confocal microscope were denoised by NIS-Elements (Nikon)'s Denoise.ai, and 3D-SIM images were reconstructed using NIS-Elements (Nikon) software.

## Transmission electron microscopy

Suspended cells were collected at the indicated time points and isolated by centrifugation. The cell pellets were sequentially fixed with 3% glutaraldehyde and with 1% osmium tetroxide on ice for 1 hr, washed with 0.1 M cacodylate buffer (pH 7.2) containing 0.1% $CaCl_2$, and dehydrated in a series of ethanol and propylene oxide solutions. The samples were then embedded in Epon 812 mixture and incubated in an oven at 60°C for 36 hr. Ultrathin sections (thickness of 70 nm) were cut with an ULTRACUT UC7 ultramicrotome (Leica) and mounted on 75-mesh copper grids. Sections

were counterstained with uranyl acetate for 10 min and lead citrate for 7 min and then examined with a KBSI Bio-High Voltage EM system (JEM-1400 Plus at 120 kV and JEM-1000BEF at 1000 kV, JEOL).

## Statistical analysis

Data are presented as means ± SD and were compared between or among groups with the two-tailed unpaired *t*-test or by one-way ANOVA with the use of Prism 9 software. A p-value of <0.05 was considered statistically significant.

## Acknowledgements

We thank Dr. Won Do Heo (Department of Biological Sciences, Korea Advanced Institute of Science and Technology) for sharing the plasmids expressing mCherry-H2B and membrane-targeted EGFP. We thank all members of the Lim laboratory for discussion. This study was supported by Samsung Science and Technology Foundation (project number SSTF-BA2001-12 to D-SL) as well as by grants (2020R1A3B2079551 to D-SL, 2022M3H9A2083956 to S-HL) from the National Research Foundation of Korea funded by the Ministry of Science and ICT.

## Additional information

### Funding

| Funder | Grant reference number | Author |
| --- | --- | --- |
| National Research Foundation of Korea | 2020R1A3B2079551 | Jeongsik Kim Dahyun Kim Dae-Sik Lim |
| National Research Foundation of Korea | 2022M3H9A2083956 | Sang-Hee Lee |
| Samsung Science and Technology Foundation | SSTF-BA2001-12 | Jeongsik Kim Dahyun Kim Dae-Sik Lim |

The funders had no role in study design, data collection and interpretation, or the decision to submit the work for publication.

### Author contributions

Jeongsik Kim, Conceptualization, Data curation, Formal analysis, Validation, Investigation, Visualization, Methodology, Writing – original draft; Dahyun Kim, Data curation, Formal analysis, Validation, Investigation, Visualization, Writing – review and editing; Dong-Kyun Kim, Resources, Formal analysis, Methodology; Sang-Hee Lee, Resources, Formal analysis, Investigation, Methodology; Wonyul Jang, Conceptualization, Data curation, Formal analysis, Supervision, Investigation, Writing – original draft, Writing – review and editing; Dae-Sik Lim, Conceptualization, Supervision, Funding acquisition, Writing – original draft, Writing – review and editing

### Author ORCIDs

Jeongsik Kim https://orcid.org/0000-0002-7503-4980
Dahyun Kim https://orcid.org/0000-0003-1859-9114
Wonyul Jang http://orcid.org/0000-0003-0352-3862
Dae-Sik Lim https://orcid.org/0000-0003-2356-7555

Reviewer #1 (Public review): https://doi.org/10.7554/eLife.96178.3.sa1
Reviewer #2 (Public review): https://doi.org/10.7554/eLife.96178.3.sa2
Reviewer #3 (Public review): https://doi.org/10.7554/eLife.96178.3.sa3
Author response https://doi.org/10.7554/eLife.96178.3.sa4

## Additional files

### Supplementary files
• MDAR checklist

### Data availability
All data produced or analyzed in this study are provided within the manuscript and supplementary materials. Original blot images and quantifications data are contained in the source files.

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
