## [Editor Report · eLife assessment]

This **important** study reports the formation of a new organelle, called giant unilocular vacuole (GUVac), in mammary epithelial cells through a macropinocytosis-like process. The evidence supporting conclusions is **convincing**, using state-of-the-art cell biology techniques. This work will be of interest to cell biologists and contribute to the understanding of cell survival mechanisms against anoikis.

---

## [Referee Report · Reviewer #1 (Public review)]

The authors found that the loss of cell-ECM adhesion leads to the formation of giant monocular vacuoles in mammary epithelial cells. This process takes place in a macropinocytosis-like process and involves PI3 kinase. They further identified dynamin and septin as essential machinery for this process. Interestingly, this process is reversible and appears to protect cells from cell death.

Strengths: The data are clean and convincing to support the conclusions. The analysis is comprehensive, using multiple approaches such as SIM and TEM. The discussion on lactation is plausible and interesting.

Weaknesses: As the first paper describing this phenomenon, it is adequate. However, the elucidation of the molecular mechanisms is not as exciting as it does not describe anything new. It is hoped that novel mechanisms will be elucidated in the future. Especially the molecules involved in the reversing process could be quite interesting.

---

## [Referee Report · Reviewer #2 (Public review)]

Summary:

The manuscript describes an interesting observation and provides initial steps towards understanding the underlying molecular mechanism.

The manuscript describes that the majority of non-tumorigenic mammary gland epithelial cells (MCF-10A) in suspension initiate entosis. A smaller fraction of cells form a single giant unilocular vacuole (hereafter referred to as a GUVac). GUVac appeared to be empty and did not contain invading (entotic) cells. The formation of GUVac could be promoted by disrupting actin polymerisation with LatB and CytoD. The formation of GUVacs correlated with resistance to anoikis. GUVac formation was detected in several other epithelial cells from secretory tissues.

The authors then use electron microscopy and super-resolution imaging to describe the biogenesis of GUVac. They find that GUVac formation is initiated by a micropinocytosis-like phenomenon (that is independent of actin polymerisation). This process leads to the formation of large plasma membrane invaginations, that pinch off from the PM to form larger vesicles that fuse with each other into GUVacs.

Inhibition of actin polymerisation in suspended MCF-10a leads to the recruitment of Septin 6 to the PM via its amphipathic helix. Treatment with FCF (a septin polymerisation inhibitor) blocked GUVac biogenesis, as did pharmacological inhibition of dynamin-mediated membrane fission. The fusion of these vesicles in GUVacs required (perhaps not surprisingly) PI3P.

Strengths:

The authors have made an interesting and potentially important observation. They describe the formation of an endo-lysosomal organelle (a giant unilocular vacuole - GUVac) in suspended epithelial cells and correlate the formation of GUVacs with resistance to aniokis.

Comments on revised version:

Additional experiments, including a better characterization of GUVac biogenesis, as well as knockdown and knock out of class II PI3Kα (PI3K-C2α) or class III PI3K (VPS34), have improved the manuscript.

---

## [Referee Report · Reviewer #3 (Public review)]

Summary:

Loss of cell attachment to extracellular matrix (ECM) triggers aniokis (a type of programmed cell death), and resistance to aniokis plays a role in cancer development. However, mechanisms underlying anoikis resistance, and the precise role of F-actin, are not fully known.

Here authors describe the formation of a new organelle, giant unilocular vacuole (GUVac), in cells whose F-actin is disrupted during loss of matrix attachment. GUVac formation (diameter >500 nm) resulted from a previously unrecognised macropinocytosis-like process, characterized by inwardly curved micron-sized plasma membrane invaginations, dependent on F-actin depolymerization, septin recruitment and PI(3)P. Finally, the authors show GUVac formation after loss of matrix attachment promotes resistance to anoikis.

From these results, authors conclude that GUVac formation promotes cell survival in environments where F-actin is disrupted and conditions of cell stress.

Strengths:

The manuscript is clear and well-written, figures are all presented at a very high level.

A variety of cutting edge cell biology techniques (eg time-lapse imaging, EM, super-resolution microscopy) are used to study the role of cytoskeleton in GUVac formation, discovering (i) a macropinocytosis-like process dependent on F-actin depolymerisation, SEPT6 recruitment and PI(3)P contributes to GUVac formation, and (ii) GUVac formation is associated with resistance to cell death.

Experimental work was advanced in response to reviewers' comments, improving the manuscript message and mechanistic advance.

Weaknesses:

The manuscript is highly reliant on the use of drugs, or combinations of drugs, for long periods of time (6hr, 18hr). However, in the revised manuscript, authors test conclusions drawn from experiments involving drugs using other canonical cell biology approaches.

The molecular characterisation of GUVacs has been advanced, although not fully resolved.

The authors show (mostly using pharmacological inhibition) that F-actin is key for GUVac formation. The precise role of F-actin / GUVac formation in anoikis resistance will be the focus of future work.

---

## [Author Response]

The following is the authors’ response to the original reviews.

**Reviewer #1 (Public Review):**
Summary:The authors found that the loss of cell-ECM adhesion leads to the formation of giant monocular vacuoles in mammary epithelial cells. This process takes place in a macropinocytosis-like process and involves PI3 kinase. They further identified dynamin and septin as essential machinery for this process. Interestingly, this process is reversible and appears to protect cells from cell death.Strengths:The data are clean and convincing to support the conclusions. The analysis is comprehensive, using multiple approaches such as SIM and TEM. The discussion on lactation is plausible and interesting.

We thank the reviewer for the summary of our study and the positive comment.

Weaknesses:As the first paper describing this phenomenon, it is adequate. However, the elucidation of the molecular mechanisms is not as exciting as it does not describe anything new. It is hoped that novel mechanisms will be elucidated in the future. In particular, the molecules involved in the reversing process could be quite interesting.

We agree with the reviewer’s comments and believe that investigating the molecular mechanisms involved in reversing GUVac formation, as illustrated in Figure 5J, would be valuable for future research.

Additionally, the relationship to conventional endocytic compartments, such as early and late endosomes, is not analyzed.

We thank the reviewer for the valuable comment. To determine whether GUVac displays markers of other endomembrane systems, we analyzed several markers, including EEA1, Rab5, LC3B, LAMP1, and Transferrin receptor (TfR). At early time points (1 h), we observed several large vesicles that had taken up 70kDa Dextran and exhibited EEA1 or Rab5, markers of early endosomes. By 6 hours, some of these large vesicles showed lysotracker positivity, indicating a transition from early to late endosomal fate, similar to the maturation process of conventional macropinocytic vesicles (see new Figure 1-figure supplement 2A). However, once the vesicles fused, grew, and became GUVac, these markers did not consistently correspond with the GUVac membrane but were instead unevenly distributed around it (new Figure 1-figure supplement 2B, C). This made it difficult to determine whether they were localized to separate organelles or part of the GUVac membrane. Interestingly, we found that the Transferrin receptor (TfR), which also marks a general membrane population involved in the endocytic pathway (such as PM invagination), was evenly distributed within the GUVac membrane (new Figure 1-figure supplement 2B, D). Therefore, GUVac appears to possess heterogeneous characteristics of the endocytic membrane, mainly with the TfR marker (likely due to PM invagination) and some partial endomembrane system markers. However, further analysis would be required to confirm this.

**Reviewer #2 (Public Review):**
Summary:The manuscript "Formation of a giant unilocular vacuole via macropinocytosis-like process confers anoikis resistance" describes an interesting observation and provides initial steps towards understanding the underlying molecular mechanism.The manuscript describes that the majority of non-tumorigenic mammary gland epithelial cells (MCF-10A) in suspension initiate entosis. A smaller fraction of cells forms a single giant unilocular vacuole (hereafter referred to as a GUVac). GUVac appeared to be empty and did not contain invading (entotic) cells. The formation of GUVac could be promoted by disrupting actin polymerisation with LatB and CytoD. The formation of GUVacs correlated with resistance to anoikis. GUVac formation was detected in several other epithelial cells from secretory tissues.The authors then use electron microscopy and super-resolution imaging to describe the biogenesis of GUVac. They find that GUVac formation is initiated by a micropinocytosis-like phenomenon (that is independent of actin polymerisation). This process leads to the formation of large plasma membrane invaginations, that pinch off from the PM to form larger vesicles that fuse with each other into GUVacs.Inhibition of actin polymerisation in suspended MCF-10a leads to the recruitment of Septin 6 to the PM via its amphipathic helix. Treatment with FCF (a septin polymerisation inhibitor) blocked GUVac biogenesis, as did pharmacological inhibition of dynamin-mediated membrane fission. The fusion of these vesicles in GUVacs required (perhaps not surprisingly) PI3P.Strengths:The authors have made an interesting and potentially important observation. They describe the formation of an endo-lysosomal organelle (a giant unilocular vacuole - GUVac) in suspended epithelial cells and correlate the formation of GUVacs with resistance to aniokis.

We thank the reviewer for the summary of our study and the positive comment.

Weaknesses:My major concern is the experimental strategy that is used throughout the paper to induce and study the formation GUVac. Almost every experiment is conducted in suspended cells that were treated with actin depolymerising drugs (e.g. LatB) and thus almost all key conclusions are based on the results of these experiments. I only have a few suggestions that would improve these experiments or change their outcome and interpretation. Yet, I believe it is essential to identify the endogenous pathway leading to the actin depolymerisation that drives the formation of GUVacs in detached epithelial cells (or alternatively to figure out how it is suppressed in most detached cells). A first step in that direction would be to investigate the polymerization status of actin in MCF-10a cells that 'spontaneously' form GUVacs and to test if these cells also become resistant to anoikis.

We thank the reviewer for the valuable comments and fully acknowledge the limitations of our approach. Many detached cells likely tend to contact each other for cell aggregations to suppress GUVac formation. However, it is unclear whether cells that spontaneously form GUVac in suspension have a weakened F-actin structure, which would be valuable to investigate in future studies.

Also, it would be great (and I believe reasonably easy) to better characterise molecular markers of GUVacs (LAMP's, Rab's, Cathepsins, etc....) to discriminate them from other endosomal organelles

In response to a similar comment from Reviewer 1, we analyzed markers of other endocytic compartments, including EEA1, Rab5, Transferrin receptor (TfR), LC3B, and LAMP1. At early time points (1 h), we observed several large vesicles that had taken up 70kDa Dextran and exhibited EEA1 or Rab5, markers of early endosomes. By 6 hours, some of these large vesicles showed lysotracker positivity, indicating a transition from early to late endosomal fate, similar to the maturation process of conventional macropinocytic vesicles (see new Figure 1-figure supplement 2A). However, once the vesicles fused, grew, and became GUVac, these markers did not consistently correspond with the GUVac membrane but were instead unevenly distributed around it (new Figure 1-figure supplement 2B, C). This made it difficult to determine whether they were localized to separate organelles or part of the GUVac membrane. Interestingly, we found that the Transferrin receptor (TfR), which also marks a general membrane population involved in the endocytic pathway (such as PM invagination), was evenly distributed within the GUVac membrane (new Figure 1-figure supplement 2B, D). Therefore, GUVac appears to possess heterogeneous characteristics of the endocytic membrane, mainly with the TfR marker (likely due to PM invagination) and some partial endomembrane system markers. However, further analysis would be required to confirm this.

**Reviewer #3 (Public Review):**
Summary:Loss of cell attachment to extracellular matrix (ECM) triggers aniokis (a type of programmed cell death), and resistance to aniokis plays a role in cancer development. However, mechanisms underlying anoikis resistance, and the precise role of F-actin, are not fully known.Here the authors describe the formation of a new organelle, giant unilocular vacuole (GUVac), in cells whose F-actin is disrupted during loss of matrix attachment. GUVac formation (diameter >500 nm) resulted from a previously unrecognised macropinocytosis-like process, characterized by inwardly curved micron-sized plasma membrane invaginations, dependent on F-actin depolymerization, septin recruitment, and PI(3)P. Finally, the authors show GUVac formation after loss of matrix attachment promotes resistance to anoikis.From these results, the authors conclude that GUVac formation promotes cell survival in environments where F-actin is disrupted and conditions of cell stress.Strengths:The manuscript is clear and well-written, figures are all presented at a very high level.A variety of cutting-edge cell biology techniques (eg time-lapse imaging, EM, super-resolution microscopy) are used to study the role of the cytoskeleton in GUVac formation. It is discovered that: (i) a macropinocytosis-like process dependent on F-actin depolymerisation, SEPT6 recruitment, and PI(3)P contributes to GUVac formation, and (ii) GUVac formation is associated with resistance to cell death.

We thank the reviewer for the concise summary of our study and positive comments.

Weaknesses:The manuscript is highly reliant on the use of drugs, or combinations of drugs, for long periods of time (6hr, 18hr..). Wherever possible the authors should test conclusions drawn from experiments involving drugs also using other canonical cell biology approaches (eg siRNA, Crispr). Although suggestive as a first approach, it is not reliable to draw conclusions from experiments where only drug combinations are being advanced (eg LatB + FCF).

We thank the reviewer for the comment and suggestion. As suggested, we employed siRNAs targeting Septin2 and Septin9 in cells treated with LatB as an alternative to the drug combination approach. This genetic approach, combined with chemical treatment, led to a consistent reduction in GUVac formation, similar to the results observed with LatB+FCF treatment (see new Figure 3D-WB and graph).

F-actin is well known to play a wide variety of roles in cell death and other canonical cell death pathways (PMID: 26292640). The authors show using pharmacological inhibition that F-actin is key for GUVac formation. However, especially when testing for physiological relevance, how can these other roles for F-actin be ruled out?

In Figure 5, we investigate the physiological relevance of GUVac, highlighting its role in suppressing apoptosis and enhancing anoikis resistance. As the reviewer correctly noted, F-actin inhibition is known to reduce apoptotic signaling (PMID: 16072039). However, we observed that anoikis resistance is lost when GUVac is suppressed through knockout of either PI3KC2alpha or VPS34 in cells with F-actin disrupted by LatB (Figure 5I). This suggests that GUVac plays a role in suppressing apoptosis independently of F-actin depolymerization-induced apoptosis resistance.

To test the role of septins in GUVac formation only recruitment studies and no direct functional work is performed. A drug forchlofeneuron (FCF) is used, but this is well known to have off-target effects (PMID: 27473917).

We thank the reviewer for the valuable comments. To eliminate potential off-target effects of FCF, as described above, we employed siRNA targeting Septin 2 and Septin 9 and observed similar results (see new Figure 3D).

Cells that possess GUVac are resistant to aniokis, but how are these cells resistant? This report is focused on mechanisms underlying GUVac formation and does not directly test for mechanisms underlying aniokis resistance.

We fully agree with the reviewer’s comments and recognize the importance of uncovering the mechanism behind GUVac-mediated anoikis resistance for future research. It will likely be essential to investigate how prosurvival signaling pathways are activated, like the PI3K-AKT signaling (as shown in Figure 5-Supplement 1) or the YAP/TAZ pathway.

**Reviewer #1 (Recommendations For The Authors):**
Figure 4 Supplemental 1. What are the faint bands in clones 23, 26, and 29? Are they cross-reacting bands? Or Vps34?

We apologize if the data in our original manuscript were misleading. To clarify the specificity of the VPS34 antibody in the Western blot analysis of VPS34 KO clones, we compared these samples with those from siRNA-mediated VPS34-depleted cells (see new Figure 4-Supplement 1E, which replaces the original Figure). Consistent with the known size of VPS34 at approximately 100 kDa, we observed a clear disappearance of the VPS34 band at around 100 kDa in the sgVPS34 clones, which was comparable to the size observed in siRNA-treated cells.

**Reviewer #2 (Recommendations For The Authors):**
Figure 2B: Only 4 cells were counted. Please comment.

At the outset of this study, we faced technical difficulties in preparing TEM samples, which limited the number of samples included in Figure 2B. However, subsequent experiments that combined TEM with super-resolution microscopy, as shown in Figure 4D-F, produced similar data on plasma membrane invagination, as depicted in Figure 2B, which is the initial step in the formation of GUVac.

Figure 2C: do cells shrink after treatment with EIPA or LatB? Please comment.

We apologize if the data presented in our original manuscript were misleading. Control cells treated with DMSO display multiple cell-in-cell structures (known as 'entosis'), which typically results in a larger overall cell size compared to EIPA or LatB-treated non-entotic single cells. This might have created the impression that cells shrink relative to the control under EIPA or LatB treatment. We hope this explanation has answered the reviewer’s question.

Figure 3A: The changes in the localization of mCherry-Spetin6 appear to be very dramatic. Are these results properly reflected by the quantification in Figure 3B? Is indeed the entire mCherry-Spetin6 pool recruited to the plasma membrane? Wouldn't that imply that all other septin6-regulated processes are blocked?

Again, we apologize if the data presented in our original manuscript caused any confusion. In Figure 3B, we quantified only the number of filament-like Septin6 structures predominantly observed in LatB-treated cells, rather than measuring changes in the relative fluorescence intensity of Septin6 between the plasma membrane and the cytosol. Although we could not estimate the proportion of total Septin6 recruited to the plasma membrane from the cytosol based solely on Figure 3A-B, conducting plasma membrane fractionation experiments with endogenous Septin6, followed by Western blot analysis, would be valuable for addressing this issue in future studies.

Figure 3D: Please also provide data for the 6h time-point (as in all other experiments).

We apologize for omitting the 6-hour time point, which may have caused confusion. The new Figure 3E (previously Figure 3D) shows that recruitment of wild-type Septin6, but not the amphipathic helix (AH) deletion mutant, occurs at a 6-hour time point.

Figure 3E: Molecular weight for western blot is missing.

We thank the reviewer for pointing this out and have revised the figure accordingly.

Line 188 - Title of subchapter could include dynamin.

We appreciate the reviewer’s helpful suggestion and have updated the revised manuscript to reflect this. The phrase "Recruitment of Septin to the Fluctuating Plasma Membrane Drives Macropinocytosis-like Process" has been revised to "Septin and Dynamin Drive Macropinocytosis-like Process".

Line 450 - please describe how the genotyping of MCF10a gene-engineered cells was performed.

We confirmed the knockout of MCF10A cell lines by Western blot analysis using specific antibodies against VPS34 and PI3KC2α, rather than through genotyping.

**Reviewer #3 (Recommendations For The Authors):**
(1) The manuscript is highly reliant on the use of drugs, or combinations of drugs, for long periods of time (6hr, 18hr..). Wherever possible authors should test conclusions drawn from experiments involving drugs also using other canonical cell biology approaches (eg siRNA, Crispr). Although suggestive as a first approach, it is not reliable to draw conclusions from experiments where only drug combinations are being advanced (eg LatB + FCF).

We thank the reviewer for the comment. As suggested, we employed siRNAs targeting Septin2 and Septin9 in cells treated with LatB as an alternative to the drug combination approach. This genetic approach, combined with chemical treatment, led to a consistent reduction in GUVac formation, similar to the results observed with LatB+FCF treatment (see new Figure 3D-WB and graph).

(2) SEPT6 is recruited at an inwardly curved plasma membrane. Can the authors better describe what type of structure is being recruited/quantified (filaments, collar-like structures, etc)?

We apologize if the data presented was unclear. As outlined in the Methods section in the original manuscript, we detected puncta-like Septin6 structures using the Find Maxima tool in ImageJ, which could include both filamentous and collar-like structures that were less apparent in the DMSO control. We have added additional explanations in the revised manuscript in the legend of Figure 3B to clarify the recruitment of Septin6.

Previous work has shown that octameric septin complexes are linking actin to the plasma membrane (PMID: 36562751). Tests for the recruitment/function of other key septins such as SEPT7 and SEPT9 to support conclusions.

As previously mentioned, to further explore the role of other septin family members in GUVac formation, we tested the roles of Septin9 and Septin2 using siRNAs and found that they are essential for this process (see new Figure 3D). Unfortunately, we were unable to assess the localization of Septin2 and Septin9 due to the lack of suitable antibodies for detecting endogenous proteins by immunofluorescence.

(3) SEPT6 recruitment is impaired when cells are treated with FCF. FCF is well known to have off-target effects (PMID: 25217460, PMID: 27473917). siRNA for SEPT2, SEPT7 and/or SEPT9 can be used to test phenotypes obtained using FCF.

We thank the reviewer for the comment. As also mentioned above, to eliminate potential off-target effects of FCF, we used siRNA to target Septin2 and Septin9, and obtained similar results (see new Figure 3D).

(4) SEPT6 is recruited to the fluctuating cell membrane via the amphipathic helix (AH) domain (Figure 3D). Are these only representative images? It is not clear what readers should be looking at - can the authors provide arrows to highlight what is the difference +/- AH? Can something be quantified?

We thank the reviewer for the suggestion and have added arrows from the inset of the merge pannel Figure 3E, along with line profile analysis, to emphasize the failure of the AH deletion mutant of Septin6 to recruit to the plasma membrane.

Throughout Figure 3, why use LatB treatment at different times?

We apologize if this was not clearly addressed in our original manuscript. Throughout the study, we primarily used an 18-hour LatB treatment to evaluate GUVac formation, as this longer period allows for gradual vesicle fusion. In contrast, we utilized 6-hour treatments to demonstrate that Septin6 recruitment and subsequent plasma membrane invagination occur at earlier time points, as evidenced by the data in Figure 2G (super-resolution live imaging) and Figure 4D (electron microscopy analysis). This clarification has been incorporated into the revised manuscript.

(5) F-actin is well known to play a wide variety of roles in cell death and other canonical cell death pathways (PMID: 26292640). The authors show using pharmacological inhibition that F-actin is key for GUVac formation. However, especially when testing for physiological relevance, how can these other roles for F-actin be ruled out?

In Figure 5, we investigate the physiological relevance of GUVac, highlighting its role in suppressing apoptosis and enhancing anoikis resistance. As the reviewer correctly noted, F-actin inhibition is known to reduce apoptotic signaling (PMID: 16072039). However, when GUVac is suppressed through knockout of either PI3KC2alpha or VPS34 in cells with F-actin disrupted by LatB, anoikis resistance is lost (see Figure 5H, I). This suggests that GUVac plays a role in suppressing apoptosis independently of F-actin depolymerization-induced apoptosis resistance.

(6) Cells that possess GUVac are resistant to aniokis, but how are these cells resistant? This report is focused on mechanisms underlying GUVac formation and does not directly test for mechanisms underlying aniokis resistance.

We fully agree with the reviewer’s comments and recognize the importance of uncovering the mechanism behind GUVac-mediated anoikis resistance for future research. It will likely be essential to investigate how prosurvival signaling pathways are activated, like the PI3K-AKT signaling (as shown in Figure 5-Supplement 1) or the YAP/TAZ pathway.

(7) In the Discussion, there is a lot of text on involution and speculative relevance of GUVac formation. I would focus the Discussion more on the clear results discovered here.

We thank the reviewer’s feedback and have revised the discussion to reduce its length concerning involution.

(8) Figure 5. GUVac formation promotes cell survival in altered actin and matrix environments. In Figure 5J, it will not be clear to readers outside the field what is being shown here.

We appreciate the reviewer’s suggestion and have added two distinct dotted lines around the vacuole and cell area in the revised figure to emphasize the gradual reduction in its size over time.